# Impaired expression of metallothioneins contributes to allergen-induced inflammation in patients with atopic dermatitis

Sofia Sirvent [1,9], Andres F. Vallejo [1,9], Emma Corden[2], Ying Teo[1,2], James Davies[1,3], Kalum Clayton [1], Eleanor G. Seaby[4], Chester Lai[1,2], Sarah Ennis [4], Rfeef Alyami[1], Gemma Douilhet[1], Lareb S. N. Dean [1], Matthew Loxham [1], Sarah Horswill[2], Eugene Healy [1,2], Graham Roberts[2,4], Nigel J. Hall[2,5], Peter S. Friedmann[1], Harinder Singh [6], Clare L. Bennett [3], Michael R Ardern-Jones [1,2,7] & Marta E. Polak [1,7,8] ✉

Regulation of cutaneous immunity is severely compromised in inflammatory skin disease. To investigate the molecular crosstalk underpinning tolerance versus inflammation in atopic dermatitis, we utilise a human in vivo allergen challenge study, exposing atopic dermatitis patients to house dust mite. Here we analyse transcriptional programmes at the population and single cell levels in parallel with immunophenotyping of cutaneous immunocytes revealed a distinct dichotomy in atopic dermatitis patient responsiveness to house dust mite challenge. Our study shows that reactivity to house dust mite was associated with high basal levels of TNF-expressing cutaneous Th17 T cells, and documents the presence of hub structures where Langerhans cells and T cells co-localised. Mechanistically, we identify expression of metallothioneins and transcriptional programmes encoding antioxidant defences across all skin cell types, that appear to protect against allergen-induced inflammation. Furthermore, single nucleotide polymorphisms in the MTIX gene are associated with patients who did not react to house dust mite, opening up possibilities for therapeutic interventions modulating metallothionein expression in atopic dermatitis.

Body surfaces such as the skin, which forms the interface with the environment, play a vital role in sensing whether environmental insults are "dangers" and communicate this to the adaptive immune system[1–5]. Immune homoeostasis is maintained in a steady state which ensures tolerance to harmless environmental insults and the inhabiting biofilm of microbiota. However, in chronic inflammatory skin conditions such as atopic dermatitis (AD), this immunotolerance is breached, resulting in uncontrolled immune responses to otherwise innocuous allergens, resulting in flares and exacerbations[6–8].

AD is one of the most prevalent inflammatory skin conditions, affecting up to 20% of children and 4–7% of adults in European countries[9,10]. Totally, 20–30% of AD cases are refractory to treatment

[1]Clinical and Experimental Sciences, Faculty of Medicine, University of Southampton, Southampton, UK. [2]University Hospital Southampton NHS Foundation Trust, Southampton, UK. [3]Department of Haematology, University College London (UCL) Cancer Institute, London WC1E 6DD, UK. [4]Human Development and Health, Faculty of Medicine, University of Southampton, Southampton, UK. [5]University Surgery Unit, Faculty of Medicine, University of Southampton, Southampton, UK. [6]Departments of Immunology and Computational and Systems Biology, The University of Pittsburgh, Pittsburgh, USA. [7]Institute for Life Sciences, University of Southampton, Southampton, UK. [8]Present address: Janssen R&D, 1400 McKean Road, Spring House, PA 19477, USA. [9]These authors contributed equally: Sofia Sirvent, Andres Vallejo. ✉e-mail: m.e.polak@soton.ac.uk

and hence very difficult to manage[11]. Importantly, even though attributing definite causes for eczematous reactions is often impossible, environmental allergens such as pollens, dust mites, pet dander from cats and dogs, moulds and human dandruff are the commonest triggers inducing allergic immune responses in eczema[8,12]. Such aberrant allergic responses are thought to be a result of complex crosstalk between an environmental trigger, impaired skin barrier, and Th2 adaptive immune activation, resulting in chronic, prolonged inflammation and uncontrolled flares[1,13–15]. But, while type 2 immunity dominates the skin of AD patients, it is overexpressed in both lesional skin and clinically non-inflamed sites[16,17], questioning the role of the Th2 immunophenotype in driving flares and acute responses to harmless allergens.

In contrast, Th17 T cells have already been shown to be of importance in AD, expressed at higher levels in the skin of children than in adults[14] and higher in intrinsic AD[15]. Our previous analysis of skin immunophenotypes indicated the existence of a specific endotype of Th17-high patients with AD, highly prevalent in chronic lesions[17].

While IL17 has been established as driving innate anti-microbial responses, TNF is a multifunctional cytokine, exerting an effect on numerous skin populations. TNF drives the expression of adhesion molecules in the skin of patients with AD, which may facilitate immune cell extravasation[18]. Together with Th2 cytokines, TNF induces AD-like features on epidermal differentiation proteins and stratum corneum lipids in human skin equivalents[19]. Induced by *Staph aureus*, TNF leads to the up-regulation of HLA-DR molecules in keratinocytes and facilitates the presentation of HDM allergen[12]. Inflammatory IL-17 and TNF-secreting CD4(+) T cells have been shown to persist even in highly immunosuppressive cancer environments[20], indicating their potential to overcome mechanisms of cutaneous homoeostasis. However, their contribution to allergen-driven responses has not yet been fully understood.

A long-standing clinical observation indicates that only a proportion of patients with detectable allergic responses on blood tests have positive eczematous responses to local skin challenges with the same allergens. While patients with severe AD show a significantly higher frequency of IgE reactivity to allergens such as cat (Fel d 1) and

house dust mites (HDM, Der p 1, 4 and 10)[21], it is not understood what determines whether such skin reactivity is present, or why, in some patients with positive blood-derived T cell reactivity to allergens, a skin challenge fails to elicit an eczematous response.

To address this question, we set up a human in vivo challenge model exposing patients with AD to a common aeroallergen, HDM and investigated transcriptional programs and the function of resident and infiltrating immune cells in reactive versus non-reactive patch test sites. This unique approach allowed us not only to delineate a network of interactions in human skin changing dynamically upon exposure to allergen but progress our understanding of molecular mechanisms safeguarding cutaneous homoeostasis. Our analysis indicates that in reactive patients, responses to HDM are mediated by Th17 TNF-expressing T cells, driving rapid expansion and overactivation of Langerhans cells (LCs). Lack of response to HDM challenge was associated with a polymorphism in MT1X linked to higher expression of metallothioneins. This network contributes to a cutaneous non-reactive state, preventing T cell activation and LC exhaustion.

## Results

### In vivo allergen challenge model to investigate mechanisms of local immune responses in human skin

To investigate the behaviour of systemic and cutaneous human immune systems upon exposure to an allergen, we set up a human in vivo allergen challenge study (Fig. 1A). We recruited 28 adult patients with moderate to severe AD under the care of a dermatologist in a tertiary referral centre. Skin barrier integrity, systemic blood responses and responsiveness to an allergen in skin prick test (SPT) were used to assess structural and systemic parameters. The study group comprised 15 men, 13 women, 89% (25/28) of Caucasian ethnicity, with median age = 37 years (IQR 24.25–53.50) (Supplementary Fig. 1A, Supplementary Data 1). Eczema severity scores (EASI) indicated moderate to severe disease (median = 17.7, IQR: 10.2–30.9, max = 51.4, Supplementary Fig. 1A, Supplementary Data 1). Skin barrier was measured as transepidermal water loss (TEWL) of non-eczematous sites and was impaired in the AD patients: median = 17.7 g/m²h, IQR: 13.3–30.7, max = 85.0 compared to healthy; median = 8.1 g/m²h, IQR: 5.9–10.8, $p < 0.0001$, Supplementary Fig. 1B). Systemic immune

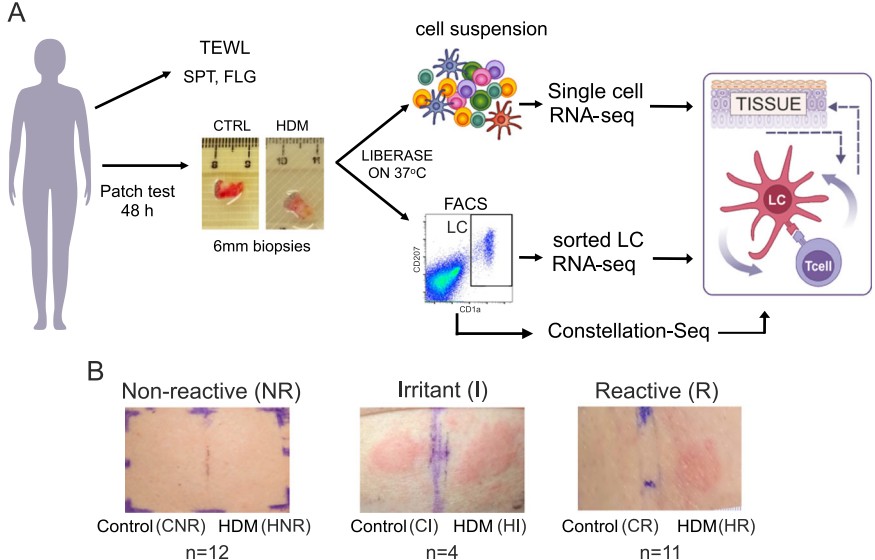

**Fig. 1 | In vivo allergen challenge model to investigate mechanisms of local immune responses in human skin. A** Human in vivo allergen challenge set-up. 6 mm biopsies taken 48 h after application of control and HDM (challenge) patch are processed to investigate transcriptional networks and regulatory interactions underpinning T cell-mediated responses to allergen. **B** Representative images of non-reactive, irritant, and reactive patch test responses to control (left) and HDM (right) allergen, 48 post-patch application. Number of patients in each group given. TEWL trans epidermal water loss, SPT skin prick test, FLG Filaggrin status, CR control patch reactive patient, HR HDM patch, reactive patient, CNR control patch, non-reactive patient, HNR HDM patch, non-reactive patient.

response to HDM was assessed using SPT. Totally, 27/28 (96%) patients showed positive SPT reactions to a range of allergens, and the reaction to HDM was one of the strongest (median wheal area 19 mm$^2$, IQR: 13–26 mm$^2$, Supplementary Fig. 1C, Supplementary Data 1). All patients had AD (diagnosed by a dermatologist as per UK Working Party diagnostic criteria[22]), and the majority suffered from hay fever (24/28, 86%) and asthma (21/28, 75%) (ISAAC questionnaire, Supplementary Fig. 1D). Local T cell-mediated responses to HDM were measured via in vivo allergen exposure patch test and assessed by the clinician. 48 hours post application of a patch test to buttock skin, 11 out of the 28 patients showed clear positive reactions to HDM and hence, were denoted as "HDM-reactive". In 12 patients, HDM did not induce a visible response. Thus they were labelled as "HDM-non-reactive" (Fig. 1B). Four patients reacted to the control patch and were thus labelled as "irritant control reactions", while 1 patient developed redness in the patch test site, which was classified by the clinician as not related to the patch test. This patient was excluded from the group analysis.

To delineate the cellular and molecular determinants underpinning local cutaneous HDM reactivity versus local tolerance, 6 mm punch biopsies were taken from control and HDM patch test sites 48 h post in vivo challenge with the allergen and cutaneous cells analysed using flow cytometry, next-generation transcriptomic and genomic sequencing.

## Reactivity to HDM is associated with the co-expansion of T cells and LCs

As expected, following HDM application in reactive patients, we observed a significant expansion of CD3 T lymphocytes ($p = 0.0001$, Fig. 2A, C), in contrast to non-reactive patch tests or irritant sites (Fig. 2B, C). This corresponded with the greater wheal areas of HDM-positive SPT in patients with HDM-reactive patch tests (Supplementary Fig. 2A, $p = 0.0109$), in agreement with observations by others[23]. Surprisingly, we observed a significant expansion of LCs (CD207+ CD1a+) and dermal DC (CD207− CD1a$^{dim}$) in skin biopsies after patch testing versus control sites (Fig. 2D–F, Supplementary Fig. 2B), in parallel to T cells.

A strong correlation between the increase in fold change of LCs and T cells compared to control ($r^2 = 0.59$, $p < 0.0001$, Fig. 2G) suggested that immune crosstalk between these cell populations perpetuates the responses to allergen. In comparison, the correlation between dermal DC and T cells was much weaker (Supplementary Fig. 2C). In situ co-localisation of LC (green) and T cells (red) was confirmed in patients reacting to HDM (Fig. 2H). Intriguingly, they created hubs akin to tertiary immune structures observed in inducible skin-associated lymphoid tissue (iSALT) previously described in a mouse model of contact dermatitis[24,25]. While these structures were less frequent in the control skin of reactive patients, T cells were localised in closer proximity to the epidermis, compared with that of non-reactive patients (Supplementary Fig. 2D).

We next tested whether the observed lack of responses to HDM in non-reactive patients with known T cell reactivity was related to a more functional skin barrier. However, the measured TEWL level indicated greater epidermal permeability in non-reactive patients ($p = 0.033$, Fig. 2I) Interestingly, high TEWL seemed to predispose to irritated control responses, perhaps highlighting that severely impaired skin barrier facilitates irritant inflammatory reactions. Furthermore, genetic analysis showed that the prevalence of variants in the filaggrin (FLG) gene, including 2282del4, R501X, S3247X, R2447X (Fig. 2J, PCR) and seven loss of function variants identified in FLG from whole exome sequencing, was comparable between HDM-reactive and non-reactive patients ($p > 0.9$, chi$^2$ test). The additional three variants (Gly1109-GlufsTer13, Ser2817AlafsTer75 and Gly323X) were of high quality (all having a genotype quality of 99 and a read depth >50). All variants had an allele balance >0.15 and did not indicate differences between reactive and non-reactive patients ($p = 0.44$, chi2 test, Supplementary Fig 2E, Supplementary Data 2). This was unsurprising given the modest

sample size is not powered to reach statistical significance. The integrity of transcriptional programming related to the epidermal barrier from reactive and non-reactive patients was further confirmed using single-cell transcriptomic data, indicating programmes encoding tight junctions, desquamation, keratinisation, cornification, lipid metabolism and desmosomes were not compromised in the skin of reactive patients (Supplementary Fig. 2F, G $n = 6$ paired biopsies, Supplementary Data 3).

In contrast to the lack of differences in skin barrier function between reactive and non-reactive patients, cutaneous CD3 +T cell infiltration in the control site was higher in reactive patients (Fig. 2K, $p = 0.0477$), indicating immune mechanisms drove responsiveness to HDM. We therefore, sought to understand in detail the molecular immune cross-talk differentiating responding and non-responding patients.

## Activated TNF-expressing Th17 cells are significantly enriched in reactive patients

To reliably track rare populations, including specific T cell types and transcriptional programmes at the level of transcription factors within dissociated skin biopsies, we applied Constellation-seq[26], a highly sensitive transcriptome read-out (Supplementary Data 4). Scanpy analysis[27] of Constellation-seq data identified 15 major cell clusters representing cell populations found in skin biopsies (Leiden algorithm, $r = 0.5$, Supplementary Fig. 3A-D, Supplementary Data 5). Cell identity was confirmed using HCA marker genes[28] (Supplementary Fig. 3E). Distinct clusters grouped keratinocytes (undifferentiated and differentiated), immune cells (including APCs and T cells), four populations of fibroblasts (F1–F4), vascular endothelial cells, lymphatic endothelial cells and pericytes and melanocytes (Supplementary Data 5).

Analysis of the T cell compartment subsetted from the Constellation Seq data identified 6 distinct T cell populations, annotated as CD4 naïve T cell, CD4 follicular helper T cells (Tfh), CD8 cytotoxic T cells, primed T cells, regulatory T cells (Tregs), and a small cluster of γδT cells, using human cell atlas (HCA) signatures[28] (Fig. 3A, B).

Cell frequency analyses confirmed observed high T cell numbers in control sites of reactive patients (Fig. 3C). These cells presented higher activation status, manifested by up-regulation of NFATC2, JUN, RELB and NFkB transcription factors (Fig. 3D). Consistently, genes up-regulated in reactive patients on exposure to HDM encoded T cell activation via JAK-STAT and Wnt signalling pathways (Fig. 3E, Supplementary Data 6). In contrast, T cells from non-reactive patients expressed transcription factors regulating tolerogenic properties (STAT5B) (Fig. 3D) and enrichment of processes of cellular senescence (Supplementary Fig. 3F), highlighting profound differences in cutaneous immune status at baseline.

Comparative analyses of the prevalence and activation of specific T cell clusters across patient groups showed that T cell changes in response to allergen were quantitative rather than qualitative (Supplementary Fig. 3G, H). Together with changes in T cell numbers observed by flow cytometry, this strongly indicated that the inflammatory process in the HDM patch test site is driven by T cell expansion from populations present in the allergen unexposed skin.

Testing for specific T cell transcriptional programmes, including Th1, Th2, Th17, Th22, and Treg (as defined by HCA), identified enrichment in Th17 cells, overrepresented across Tfh and CD4 naïve clusters. These cells, with a specific immunophenotype of CD3+ IL17+ TNF+ CD69+, were enriched at the control site of patients reactive to HDM (Fig. 3F, G). In contrast, Th1, Th2 and Th22 immunophenotypes were shared between cells across patient groups and T cell populations, resembling continuous phenotype clouds of T cells in gut tissue, as described by Kiner and colleagues[29]. Interestingly, even though Th2 determinants were expressed more strongly in patients with reactive HDM-patch tests, Th2 polarisation was also evident in non-reactive patients, both at the control site and following in vivo challenge with

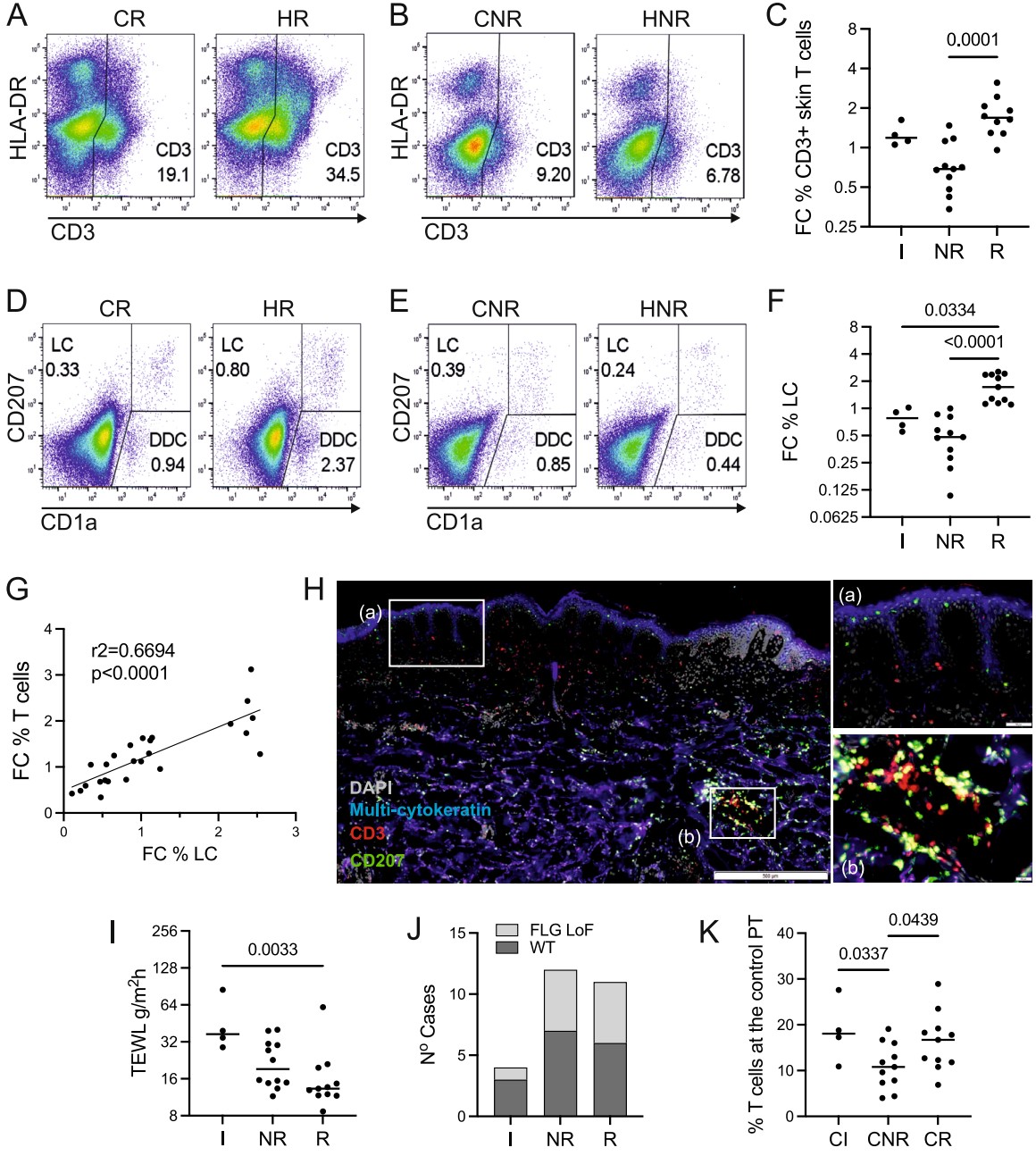

**Fig. 2 | Reactivity to HDM is associated with the co-expansion of T cells and LCs.**
**A–F** Frequency of immune cells in control and HDM patch tests from reactive vs. non-reactive patients, measured by flow cytometry. The number in the graph indicates the percentage of cells in the positive gate. CR control patch, reactive patient, HR HDM patch, reactive patient, CNR control patch, non-reactive patient, HNR HDM patch, non-reactive patient. Representative examples. **A**, **B** CD3+ T lymphocytes, **D**, **E** CD207/CD1a positive LCs. **C**, **F** Fold changes (FC) in the percentage of detected immune cells between HDM patch test and control patch test from patients with irritant, non-reactive and reactive reactions to HDM.
**G** Correlations between fold changes in the percentage of CD3+ T cells and LCs. Pearson correlation coefficient is shown. **H** Immunofluorescence staining of HDM-reactive patch test site. Inserts show the indicated optical fields at the epidermis (top) and in the dermis (bottom). Hub structures of co-localising CD207 (green) and CD3 (red) in the dermis. Epidermal layer stained with multi-cytokeratin (blue). DAPI stain for nuclei (grey). Scale bars: 500 µm, 50 µm (insets). A representative of $n = 3$ individual donors. **I** Functional assessment of skin barrier: TEWL measurements across patient groups. **J** Number of irritant (IR), non-reactive (NR) and reactive (R) cases with loss of function (LoF) variants in FLG compared to wildtype (WT). **K** Percentage of CD3+ T cells in control patch test sites identified by flow cytometry. Statistical significance was assessed by $t$-test. **C**, **G** NR $n = 11$, R $n = 10$, **F**, **K** NR $n = 11$, R $n = 11$, **I**, **J** IRR $n = 4$, NR $n = 12$, R $n = 11$. Statistical significance was assessed by the Kruskal–Wallis test with post hoc Dunn test (**C**, **F**, **I**) and unpaired ANOVA with post hoc Fisher test (**K**) following the normality Kolmogorov–Smirnov test of data distribution. Source data are provided as a Source Data file.

the allergen (Supplementary Fig. 3I, J). This is in agreement with earlier findings reporting Th2 responses in non-lesional eczema skin[16,17].

To test whether the Th17 overexpression was translated into functional protein synthesis, we assayed cytokine expression in peripheral blood mononuclear cells from reactive and non-reactive patients. IL-17-producing T cells were significantly overrepresented in the blood of reactive patients even prior to stimulation with HDM (Fig. 3G, H), while no differences were observed in IL13-producing CD4+ T cells between patient groups (Supplementary Fig. 4K, L). We next hypothesised that TNFa–Th17 cells would be in cross-talk with LCs. Indeed, CD3+ CD17+ co-localised with CD207+ LCs in dermal hubs in the control and HDM-exposed skin of HDM reactive patients (Fig. 3J,

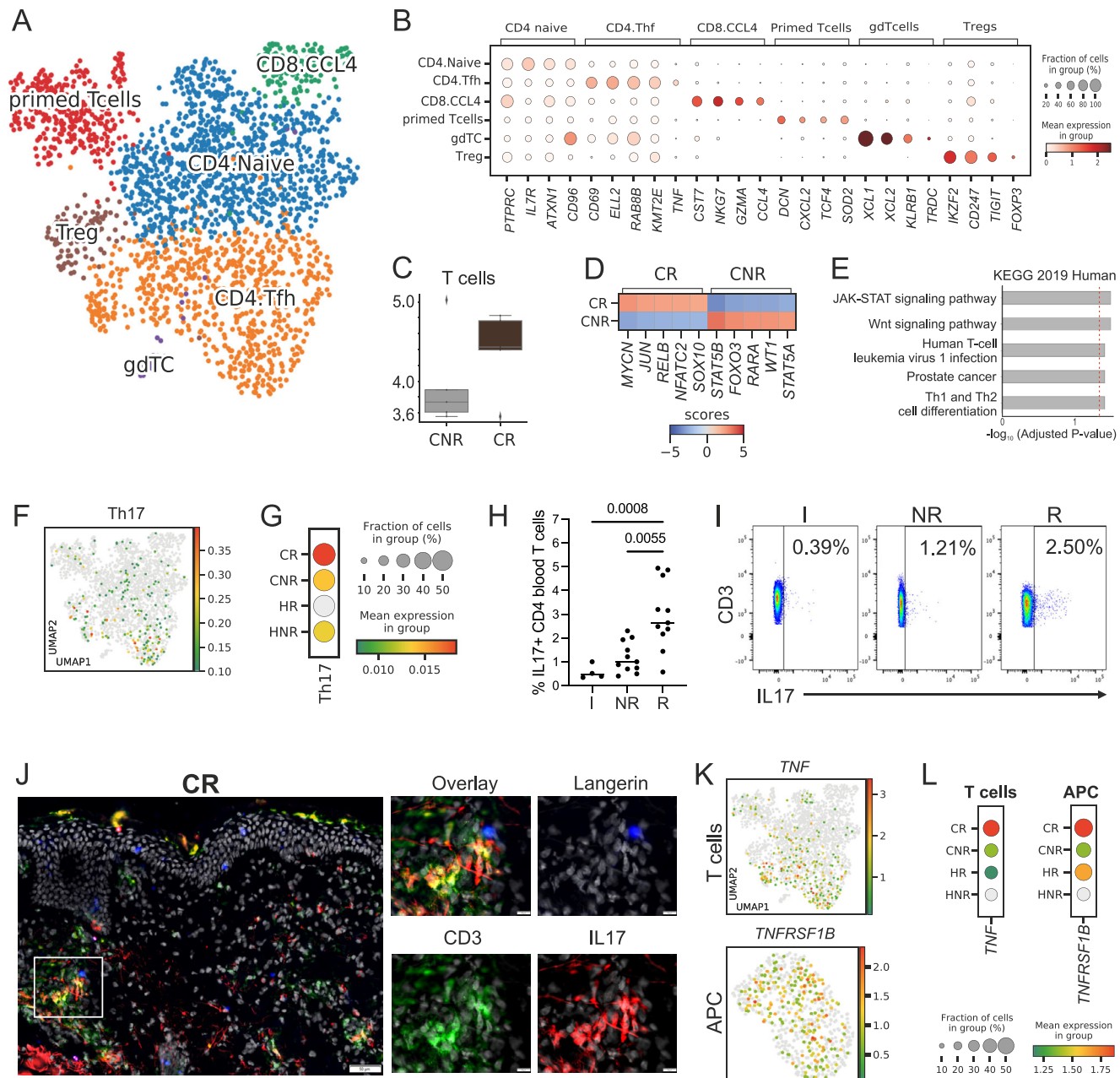

**Fig. 3 | Activated TNF-expressing Th17 cells are significantly enriched in reactive patients.** Constellation-seq analysis enriched for 1161 transcripts in 2374 single T lymphocytes cells from patch test skin biopsies, *n* = 10 patients, 5 per group. **A** UMAP plot depicting clustering of T lymphocyte populations. **B** Cell subset defining markers (Wilcoxon rank test). **C** Number of T cell transcriptomes in control reactive (CR) and non-reactive (CNR) patients. The central line denotes the median, boxes represent the interquartile range (IQR), and whiskers show the distribution except for outliers. Outliers are all points outside 1.5 times the IQR. **D** Top transcription factors expressed in T cells from reactive (CR) and non-reactive (CNR) patients in the control samples. **E** Top biological pathways enriched at the control site in DEGs from patient reactive (CR) to HDM (KEGG database), *p*-value computed using the Fisher exact test, with Benjamini Hocheberg FDR correction. **F** UMAP plots showing expression of Th17 gene signature. **G** Th17 gene signature across patient groups, dot plot: size depicts % of expressing cells, colour intensity encodes

mean expression in the group. **H** %IL17 producing CD3+ CD4+ T cells from PBMCs in irritant (I) non-reactive (NR) and reactive (R) patients. Kruskal–Wallis test with post hoc Dunn test. **I** A representative plot of IL17 expression in CD3+ CD4+ T cells I, NR and R patients. **J** Immunofluorescence staining of HDM-reactive patch test site. Inserts show the indicated optical field in the dermis. Hub structures of co-localising CD207 (blue), CD3 (green) and IL17 (red). Epidermal layer stained with multi-cytokeratin (blue). DAPI stain for nuclei (grey), Scale bars: 50 μm. Representative of n = 2. K) UMAP plots showing expression of *TNF* and *TNFRSF1B* across T cells (top) and APCs (bottom). **L** *TNF* and *TNFRSF1B* expression level across patient groups, dot plot: size depicts % of expressing cells, colour intensity encodes mean expression in the group. CR control reactive, HR HDM reactive, CNR control non-reactive, HNR HDM non-reactive. *n* = 5/group, C and H paired. Source data are provided as a Source Data file and via GEO.

Supplementary Fig. 3M). The importance of TNF for inter-cellular communication in the skin of reactive patients was next confirmed in crosstalk analyses, demonstrating a TNF:TNFSFRB signalling edge between T cells and APCs, already present in the control skin, and strengthened on exposure to HDM in reactive patients (Fig. 3K, L).

## Expression of metallothioneins counterbalances LC over-activation differentiating HDM reactive and non-reactive patients

LC is programmed in healthy skin to maintain tolerogenic networks of T cells[30,31] but may be subverted to activate pathogenic T cells in

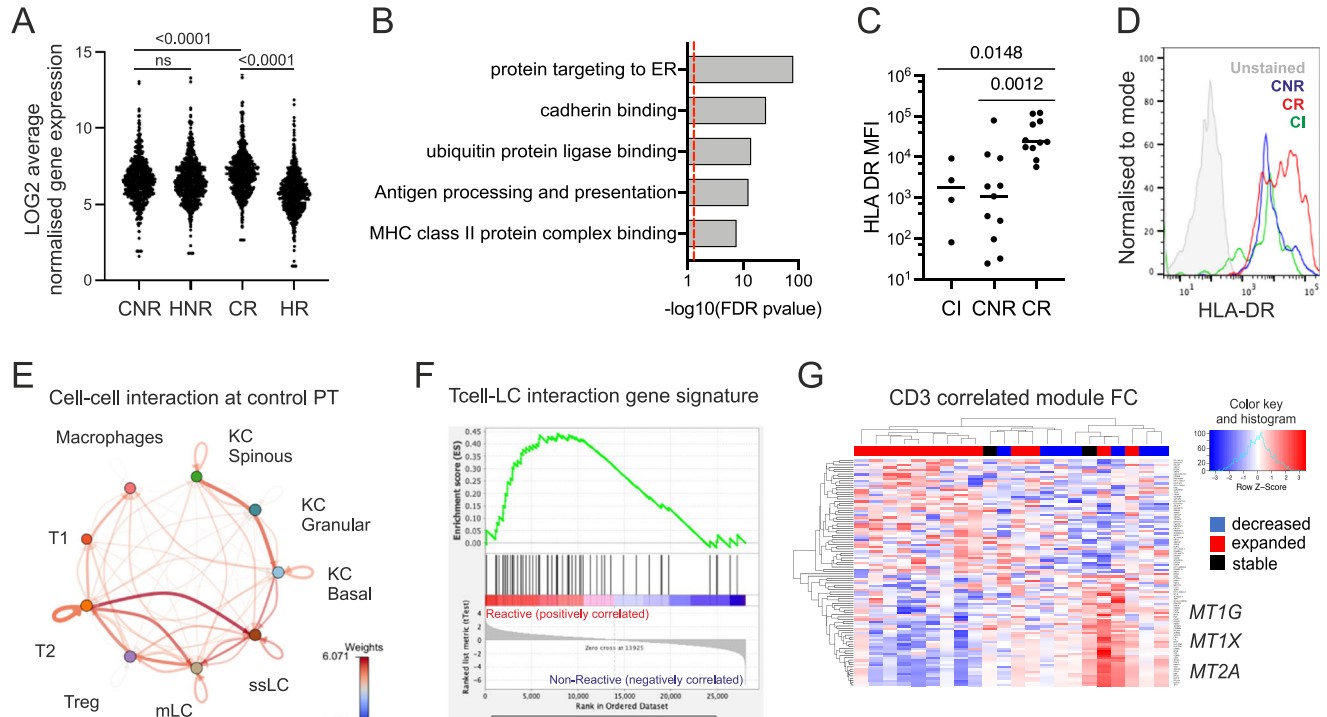

**Fig. 4 | Expression of metallothioneins counterbalance LC overactivation differentiating HDM reactive and non-reactive patients. A** Average log gene expression levels of genes in the main cluster encoding LC core programmes across non-reactive and reactive patients. Transcript to transcript clustering Biolayout, 691 genes, $r = 0.85$, MCL = 1.7. Each dot represents an average gene expression. CR: control reactive, HR HDM reactive, CNR control non-reactive, HNR HDM non-reactive, repeated measure one-way ANOVA. **B** Gene Ontologies are overrepresented in the main cluster encoding LC programmes, ToppGene, Benjamini–Hochberg-adjusted $p$-value. **C** HLA-DR expression levels measured by Flow Cytometry in CD207+ LCs. Kruskal-Wallis test with post-hoc Dunn test. D) A representative histogram of HLA-DR expression in control patch tests of CI (green), CNR(blue) and CR(red) patients compared with unstained control (grey). **E** Crosstalk analysis of interactions overexpressed between cell populations at the control site. DropSeq, $N = 6$ biopsies, 1NR, 2R patients. **F** GSEA enrichment profile in CR (red) and CNR (Blue) patients. **G** Heatmap showing segregation of patients with CD3 T cell numbers decreasing (blue), expanding (red) and stable (black) in reaction to patch test using fold change expression values of 100 top differentially regulated genes in module turquoise. Genes overexpressed in non-reactive samples contain members of the metallothionein family. CI, HI $n = 4$, CNR, HNR n = 11, CR, HNR $n = 7$ paired samples. WGCNA analysis, Pearson coefficients denoting correlation, a univariate regression model with pairwise complete Student $t$-test. Source data are provided as a Source Data file and via GEO.

disease[32]. Given the importance of TNF in APC maturation and the co-localisation of Tcell:LCs in the hubs, we hypothesised that LC function was altered in patients reacting to HDM. To test this hypothesis, we analysed transcriptional changes and molecular cross-talk in LC across experimental groups.

To investigate in-depth transcriptional changes in LCs in non-reactive vs reactive patients, CD207+ CD1a+ LCs were purified from control and HDM-challenged patch test sites (Supplementary Fig. 4A–E). Transcriptome profiles from LCs in control biopsies across patient groups were consistent with steady-state LCs described by us recently[33] (Supplementary Fig. 4F, Supplementary Data 7). Transcript-to-transcript correlation analysis of 28032 filtered and normalised transcripts (BioLayout $r = 0.85$, MCL = 1.7, minimal cluster size = 10 genes) identified 10 clusters of 1115 co-expressed genes, clearly split into two distinct structures (Supplementary Fig. 4G, H).

Core transcriptomic programmes, encoding key LC functions such as protein targeting to ER (BH adj $p = 5.67E–82$), ubiquitin proteinase ligase binding (BH adj $p = 5.83E–10$), and antigen processing and presentation (BH adj $p = 7.38E–9$), key for LC function[33], were up-regulated in LCs isolated from control patch test sites of HDM-reactive vs. non-reactive patients (Fig. 4A, B, $p < 0.0001$, Supplementary Data 8). This response was consistent with TNF-induced LC activation previously shown by us[33–35]. Consistently, flow cytometry measurement of the HLA-DR expression level further confirmed higher antigen-

presenting abilities of LCs from control patch sites of HDM-reactive patients ($p = 0.0012$, Fig. 4C, D).

To identify signals driving LC activation in the skin of reactive patients, we assessed molecular crosstalk in patch test sites. Ligand-receptor analyses of single-cell transcriptomes isolated from full skin biopsies (Supplementary Fig. 4I) confirmed that signalling from activated T lymphocytes was the strongest interaction (Fig. 4E) and delivered activation stimulus via TNF: TNFRSF1B to LC (Supplementary Fig. 4I–K). To test a hypothesis that such LC:T cell cross-talk was differentiating HDM responding and non-responding patients, we tracked the identified gene signature (Supplementary Data 9) in bulk LC transcriptomes using GSEA[36]. This confirmed that the T cell:LC signalling edge was enriched in the control sites of reactive patients (Fig. 4F, normalised enrichment score = 1.23).

Surprisingly, despite the activation of core LC functional pathways in reactive patients, our analyses demonstrated a global impairment of LC transcriptional programming after exposure to HDM, compared to both paired control patch tests and non-reactive patches. The majority of genes (691, Clusters 01, 03, 04, 05, 07, 08, 10) followed a characteristic pattern of expression, with significant downregulation after exposure to HDM in cells isolated from reactive patients (Fig. 4A, B, Supplementary Fig. 3G, H, Supplementary Data 8). Consistent with observed transcriptional changes, expression of CD207, the LC hallmark antigen uptake receptor, was decreased in LC from reactive skin patches following exposure to

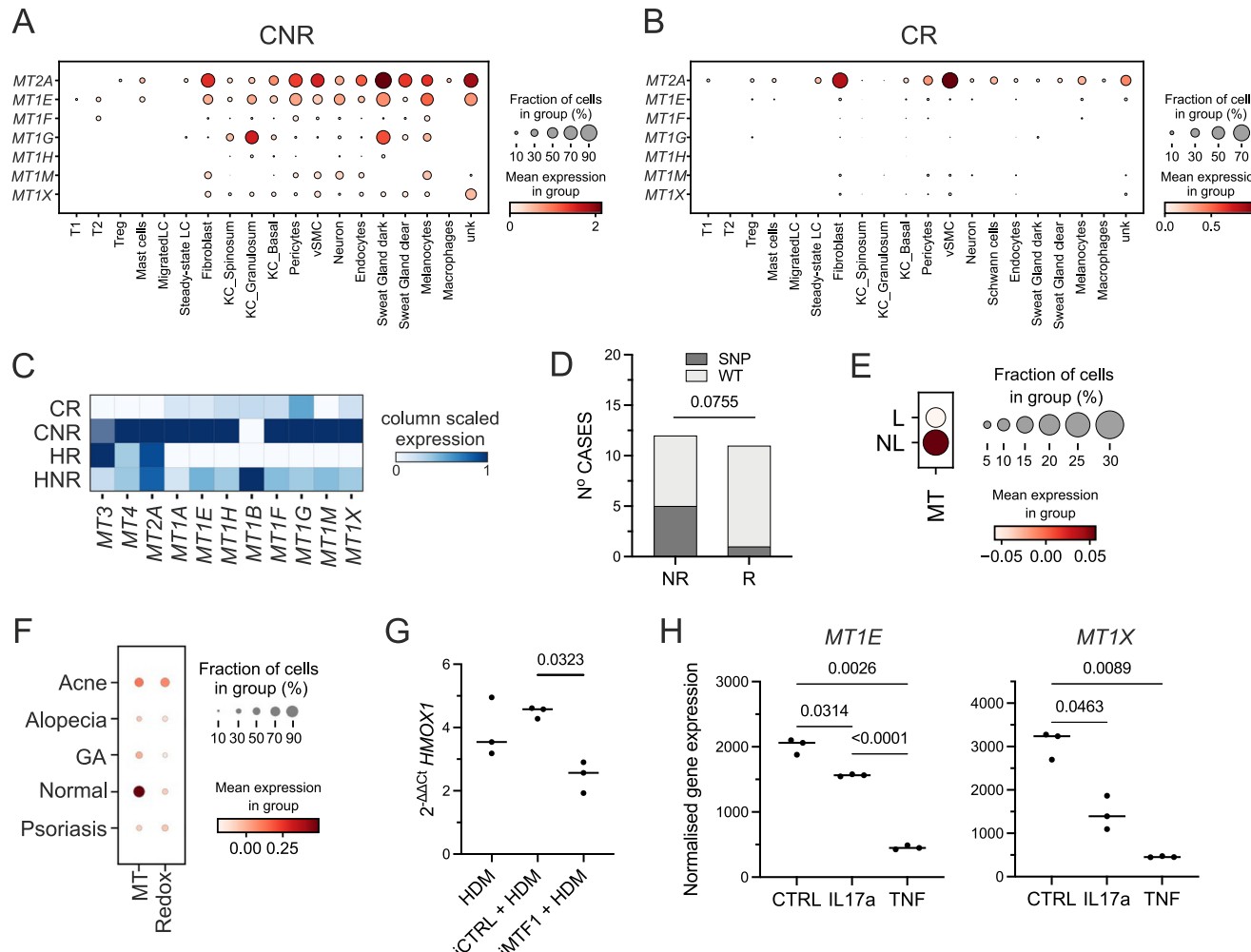

**Fig. 5 | Enhanced expression of metallothionein genes protects non-reactive patients from inflammation and prevents HDM-induced oxidative stress.**
**A, B** Dot plots showing expression of metallothioneins across cell types in control patch tests from patients with non-reactive (**A**) and reactive (**B**) patch test reactions to HDM. SCRAN-normalised single-cell RNA expression is shown for each transcript. $n = 3$, fresh skin biopsies, Drop-seq. **C** Heatmap comparing levels of expression for metallothioneins in whole skin using high sensitivity Constellation-seq method, $n = 10$ skin donors. CR control reactive, HR HDM reactive, CNR control non-reactive, HNR HDM non-reactive. **D** Distribution of WT vs SNP in MT1X gene across patient groups. Chi-square test = 3.159, two-sided, d$f$ = 1. **E** Expression of metallothioneins in patients with AD, in lesional (L) and non-lesional (NL) skin. **F** Expression of signatures encoding metallothioneins (MT), and RedOX, in patients with T-cell mediated skin diseases, $Z$-score, GSE150672. **G** Effect of silencing of *MT1F* on the expression of *HMOX1* in HDM stimulated fibroblasts. $n = 3$ independent experiments, paired ANOVA with Tukey test. **H** Normalised expression levels of genes encoding *MT1E* and *MT1X* in keratinocytes exposed to IL17a and TNF. GSE36287, $n = 3$ biological replicates, paired ANOVA with Tukey test. **A**, **B**, **E**, **F** dot plot: size depict % of expressing cells, colour intensity encodes mean expression in the group. Acne acne vulgaris, Alopecia alopecia areata, GA granuloma annulare. Source data are provided as a Source Data file.

HDM compared to non-reactive and irritant skin (Supplementary Fig. 4L, M).

Unbiased weighted gene expression network analysis (WGCNA[37]) further confirmed strong correlation existed between LCs transcriptome modules, disease severity as measured by EASI (module yellow, |$r$| = 0.48, adj $p$ = 0.001), level of CD3+ T cell infiltrate in the skin (module turquoise, encoding antigen binding, MHC class II receptor activity, |$r$| = 0.31, adj $p$ = 0.04), and Th17 T cell frequency (module blue, |$r$| = 0.32, adj $p$ = 0.04) (Supplementary Fig. 3N, Supplementary Data 10).

Strikingly, LCs from samples where CD3 T cells expanded on exposure to HDM (red) were separated by reduced expression of genes in the turquoise module, including metallothioneins *MT2A*, *MT1G*, *MT1X*, ferritin chains (*FLT*, *FTH1*) and haem oxygenase (*HMOX1*) (Fig. 4H), creating a network of antioxidant defences (Supplementary Fig. 4O). This indicated that while activated T-cell signalling likely induces LC maturation, expression of metallothioneins might provide a protective mechanism in non-reactive patients.

## Enhanced expression of metallothionein genes protects non-reactive patients from inflammation and prevents HDM-induced oxidative stress

Having identified that metallothionein expression counterbalances high activation of LC, we sought to test whether metallothionein genes induce a tolerogenic or quiescent environment in the skin.

Drop-seq analyses of freshly dissociated biopsies ($n = 6$) confirmed the high expression of metallothionein gene family across cell populations in the control sites of non-reactive vs reactive patients (Fig. 5A, B, Supplementary Fig. 2F). To ensure that the detection limits and drop-outs were not masking metallothionein expression we further corroborated the results using high-sensitivity Constellation-seq (Fig. 5C, Supplementary Fig. 5A). Differential gene expression analyses in Constellation-Seq data (Supplementary Fig. 3A, E) performed for each specific cell population between sample phenotypes indicated that amongst all the cell populations the most DEGs differentiating reactive and non-reactive patients were expressed at the control site by differentiated KCs (825 DEGs up-regulated in responding patients,

445 DEGs up-regulated in non-responding patients, MAST, FDR < 0.05, Supplementary Data 11). While DEGs up-regulated in responders encoded skin differentiation (FDR = 6×10E−13), hyperkeratosis (FDR = 4×10E−3), and skin inflammation (FDR = 3×10E−3 (ToppGene, Supplementary Fig. 5B, Supplementary Data 11), anti-oxidant defences including glutathione peroxidase activity (FDR = 6.58×10E−3) and stress responses/detoxification (FDR = 3×10E−6) were strongly enriched in non-responding patients (Supplementary Data 11). Metallothioneins: *MT2A*, *MT1M* and *MT1E* were in the top 5 most differentially expressed genes. While analyses of differentially expressed genes across different skin populations identified only sporadic genes up-regulated in non-reactive patients, these differentially up-regulated genes consistently included members of the metallothionein family, with *MT2A* being the top overexpressed gene in non-reactive fibroblasts and venous endothelium (FDR = 0.015, Supplementary Fig. 5C). While expression of metallothionein transcripts was reduced on exposure to HDM in all patients, cells from biopsies non-reactive to HDM retained some expression of metallothioneins (Fig. 5C).

Given the uniform downregulation across the skin cell types, we tested whether DNA polymorphisms could underpin differences in non-reactive vs reactive patients. Hypothesis-driven GenePy logistic regression analysis[38] of 34 genes encoding oxidative stress responses and key immunological and structural features predefined in the study (Supplementary Data 13) identified *MT1X* as the top gene differentiating patient groups. A chi-square test of independence was performed to examine the relationship between single nucleotide polymorphism (SNP) in *MT1X* and reactivity to HDM. The relation between these variables was close to significant, chi$^2$ = 3.96, *p* = 0.076, indicating that the existence of an SNP in *MT1X* protected from allergen-driven inflammation (Fig. 5D). Examining the genomic region of *MT1X*, we confirmed that both SNPs ((GRCh38) 16:56682411:G > T and (GRCh38) 16:56682435:C > A) were localised in the promoter and enhancer region of *MT1X* gene (Supplementary Fig. 5D).

We next confirmed the expression of metallothionein genes was decreased in chronic AD lesions (Fig. 5E) and compromised across a range of T-cell mediated skin diseases in comparison to healthy skin (Fig. 5F). To test the regulatory role of metallothioneins in this transcriptional network we transiently silenced expression of *MTF1*, a transcription factor coordinating the expression of metallothionein family using siRNA. We confirmed that exposure to HDM induces *HMOX1* in human fibroblasts (Fig. 5G). Silencing of *MTF1* reduced both the expression of metallothioneins, and HDM-induced *HMOX1*, providing the causal link between allergen exposure, anti-oxidant responses, and the protective role of metallothioneins (Fig. 5G, Supplementary Fig. 5E, F). We next asked the question of whether cytokines produced in the skin of patients responding to HDM could impact metallothionein expression. Indeed, analysis of publicly available data of keratinocytes exposed to a range of cytokines indicated that expression of metallothioneins could be downregulated by IL17a and TNF (Fig. 5H), providing an inducible mechanism by which T cell-mediated allergic immune responses affect anti-oxidant responses, and render the skin susceptible to chronic inflammation. In summary, we document that expression of metallothioneins is linked to a non-reactive environment in the skin, supports cutaneous anti-oxidant responses, and provides a potential protective mechanism against inflammation.

## Discussion

Accurate regulation of cutaneous immunity is fundamental for human health and quality of life. Inappropriate immune activation results in inflammatory disorders, affecting up to 40% of the population[39–41]. In AD, this problem is particularly severe, manifested by frequent exacerbations resulting in significant morbidity in paediatric and adult patients[6,8,9].

Comparing cutaneous and systemic responses of eczema patients to HDM, we demonstrated that despite evident allergen-specific responses in blood, nearly 50% of patients did not react clinically to an epicutaneous patch test with the allergen. We ruled out the possibility that the lack of reactivity might be due to the epidermal permeability barrier preventing the penetration of the allergen, as these individuals had a less effective barrier, as indicated by increased TEWL. This suggested the existence of local epidermal tolerance in the non-lesional skin and posed a question about the factors regulating the distinctly different outcomes between reactive and non-reactive patients.

Our study documents that in allergen-responsive individuals, an innate state of tolerance is overcome by inflammatory signalling, epitomised in crosstalk between activated Tfh Th17 TNF-expressing T cells and LCs. Importantly, the baseline state in the skin is distinctively different in non-reactive and reactive patients. These differences are localised mainly to the epidermis and the immune compartment. The frequency and the state of activation of Tfh Th17 TNF-expressing T cells appear to be critical for subsequent reaction to the allergen. Tfh has been previously implicated in driving re-call allergic responses to HDM in lungs[42]. However, our study defines these cells as Th17 TNF-expressing, in contrast to Th2 cells reported by others.

The role of Th17 skin infiltrating T cells in driving acute responses to an allergen could be conceivably executed by contributing to an augmented state of immune readiness, promoting a more severe immune reaction. As reported extensively and captured by cross-talk analyses, TNF is a key player in cutaneous immune signalling. TNF secreted by T cells could play a double role in driving LC function by regulating both LC migration and maturation[33,34,43–45]. Expressed in unperturbed skin, it can drive LC activation. On exposure to HDM, TNF produced by activated T cells will provide a chemotactic signal for LC to migrate out of the epidermis. In the hub structures observed in HDM reactive patients, activated LCs would support T cell activation and survival, perpetuating inflammation. However, in addition to delivering chemotactic and pro-maturation signals, TNF has cytotoxic functions, likely resulting in the observed expansion and loss of functional transcriptomes of LCs. Several lines of evidence point to the importance of the ability of LCs to induce immunotolerance as critical for cutaneous homoeostasis[30,31,46]. In this context, loss of LC function could likely lead to uncontrolled inflammation in situ, as observed in the HDM reactive patch test. Such exhaustion induced by over-activation has previously been observed in dendritic cells and macrophages in chronic infection and when overwhelmed by antigen load[47,48].

Our unbiased analysis uncovers the role of anti-oxidant defences counterbalancing immune activation in non-responding patients. In contrast to sub-clinical inflammation in the skin of reactive patients, expression of metallothioneins was associated with protection from HDM-driven inflammatory reaction. Levels of expression of metallothioneins seemed to be controlled both at the constitutive (via SNP in *MT1X* gene) and inducible levels (regulated by the acute oxidative stress/inflammation), highlighting the importance of anti-oxidative defences in AD skin.

Importantly, the anti-oxidative defence was one of the transcriptomic modules critically compromised in LCs from reactive patch test sites. Oxidative stress is one of the key components driving allergic sensitisation, and in asthma models, aeroallergens, such as HDM, directly induce the production of reactive oxygen species and DNA damage and dampen antioxidant responses[49,50]. Additionally, we and others demonstrate that cytokine signalling can affect metallothionein expression in an inducible manner. Indeed, increased oxidative stress during AD exacerbation[51] and decreased antioxidant capability in children with eczema[52] have been previously observed. Since oxidative stress itself exhausts antioxidant responses and can be induced by

many eczema-associated factors, including allergens, hormones and chemicals, it is possible that chronic exposure to such triggers may compromise LC function in the epidermis of individuals with eczema making them less able to maintain cutaneous tolerance and extending our observation beyond the patch test system.

Based on our analyses, we propose a model in which sub-clinical inflammation exhausts metallothionein stores in the skin of genetically pre-disposed patients. The sub-cutaneous inflammation is in parallel manifested by infiltration of TNF-expressing activated T cells and activated antigen-presenting cells, including LCs. In response to allergens, these quickly initiate inflammatory responses and expand T cell populations driving an inflammatory reaction. This, in turn, leads to exhaustion of LCs, uncontrolled inflammation and lesion formation, thereby mediating clinical signs of inflammation. In healthy/non-inflamed skin, allergen exposure, in the absence of subclinical inflammation, mediates immunological non-responsiveness, but in the inflamed skin depleted of oxidative defences, exposure to HDM initiates allergic inflammation.

Our current study provides a detailed description of cellular and molecular crosstalk in the skin of eczema patients, proposing a mechanism supporting the development of allergen-induced inflammation. We conclude that therapeutic interventions aimed towards disrupting Th17/TNF mediated immune cross-talk or directed towards enhancing antioxidant responses, can be harnessed to improve skin health and prevent exacerbations of AD.

## Methods

The research was conducted in the UK in collaboration with the University Hospital Southampton NHS Foundation Trust, Southampton, UK. The research is locally relevant, and the study design has been consulted with patient advocacy groups. The study has been approved by South East Coast−Brighton & Sussex Research Ethics Committee.

### Study design

Informed, written consent was obtained as per approval of South East Coast−Brighton & Sussex Research Ethics Committee in adherence to Helsinki Guidelines (approval: 16/LO/0999). Adult AD patients with mild to severe disease (mean objective EASI) were recruited through the Dermatology Centre, University Hospital NHS Trust, Southampton. All AD patients fulfilled the diagnostic criteria for AD as defined by The UK Working Party[22]. Twenty-eight patients were recruited. One of the samples was excluded due to disagreement on the PT outcome (observed minimal redness, but not consistent with the patch test perimeter). FACS results for CD3 T cells were compromised for one sample due to a technical fault. One sample was processed for single-cell RNA-seq only. Additional QC inclusion criteria were applied for bioinformatic analysis. Objective EASI was measured as described previously[53]. Before sampling, patients were washed out from any immunosuppressive treatment for at least five half-lives of the drug. Atopy status was assessed for each patient using SPT to six most common allergens: HDM, grass pollen, tree pollen mix, mixed mould, cat and dog. Histamine was used as a positive control (ALK-Abello, Horsholm, Denmark). Maintenance of normal epidermal barrier function was measured by trans-epidermal water loss (TEWL). On enrolment, information about participants' demographics and previous medical history, immediate family history and information about the atopic disease (eg eczema, rhinitis) in the subject was collected based on the ISAAC questionnaire[54]. Peripheral blood mononuclear cells (PBMC) were separated from venous blood and processed for DNA extraction. FLG mutation analysis was performed as described previously[55,56]. Briefly, primer pairs were used to amplify the region of interest from DNA prepared from peripheral blood samples of individuals with AD and controls. FLG variants R501X, 2282del4, S3247X and R2447X, which covers more than 90% of FLG mutations in a UK population, were then identified using restriction enzyme digest of

PCR products with agarose gel electrophoresis and FLG variants were confirmed by whole exome sequencing.

### In vivo allergen challenge model

HDM allergen (ALK-Abello, Horsholm, Denmark AD01-AD10, Citeq Biologics, Netherlands AD11-AD28) contained in paraffin was applied via an epicutaneous patch applied to the upper buttock skin at a non-lesional site (free from eczema) following 10× tape strip procedure to remove stratum corneum, according to our previous method[1,57]. A control patch was applied in parallel following an identical procedure, except for the HDM allergen. In all AD volunteers, this site showed no evidence of active eczema, and the volunteers were not being treated with topical therapy. Clinical responses were quantified 48 hours later at each challenge site by a specialist registrar in dermatology trained in patch testing. Totally, 6 mm skin biopsies were taken under local anaesthesia from allergen-exposed and controlled skin.

### Cell isolation

6 mm biopsies were minced using a surgical scalpel and digested for 16 h at 37 C with agitation in RPMI with LiberaseTM (Roche) following the manufacturer's instructions. After 16 h of digestion, cells were collected and washed with RPMI 5% FBS. Cells were resuspended in PBS 1% bovine serum albumin (BSA) 20 mM EDTA and filtered through 70 μm sterile filters before surface antibody staining for FACS or processing for Dropseq analysis.

### Flow cytometry and cell sorting

All antibodies were used at pre-titrated, optimal concentrations. All flow cytometry was undertaken with FACS Aria flow cytometer (BD Biosciences). For surface staining of live cells, a buffer containing PBS 1% BSA was used for all antibody staining. FACS Aria flow cytometer (Becton Dickinson, USA) was used for the analysis of human LCs for the expression of CD207, CD1a, HLA-DR (mouse monoclonal antibodies, CD1a, CD207:Miltenyi Biotech, UK and HLA-DR: BD Biosciences, UK) or T cells for the expression of CD3, CD25 and CD103 (Miltenyi Biotech). Singlets (FCSA:FCSH), CD207+/CD1a+ digested LCs were sorted into trizol for RNA isolation. In parallel, LC-depleted skin cells were sorted into RPMI 5% FBS and processed for immediate cryostorage. Antibody details are listed in Supplementary Data 14.

For intracellular cytokines, freshly isolated PBMCs were activated with anti-CD3 and anti-CD28 (1 mg/μ1) with GolgiPlug (BD Biosciences, Oxford, UK). The Cytofix/Cytoperm kit (BD Biosciences) was used according to the manufacturer's instructions. Flow cytometric analysis was undertaken following lymphocyte gating on Forward/Side scatter. Subsequent gating on CD3− PerCP 5.5 (eBiosciences), CD4-VioGreen (Miltenyi Biotech) was based on appropriate negative controls to demonstrate IL13-APC and IL-17-FITC positive cells (Miltenyi Biotech). Antibody details are listed in Supplementary Data 14.

Flow cytometry data analysis was carried out with the FlowJo software (Tree Star, Ashland).

### MTF1 silencing

MRC5 lung fibroblasts were obtained from the European Collection of Authenticated Cell Cultures (ECACC). All cultures were tested and free of mycoplasma contamination. Fibroblasts were cultured in T75 flasks in Dulbecco's Modified Eagle's Medium (DMEM) supplemented with 10% foetal bovine serum (FBS), 50 units/ml penicillin, 50 μg/ml streptomycin, 2 mM L-glutamine, 1 mM sodium pyruvate and 1× non-essential amino acids (Life Technologies, Paisley, UK), at 37 °C and 5% $CO_2$.

Prior to use, cells were seeded at 80,000 cells per well in 12-well plates and reverse transfected with short interfering RNA (siRNA) against MTF1 (L-020078-00-0005; Dharmacon, UK) at a final concentration of 20 nM using Lipofectamine RNAiMAX reagent (Invitrogen) and OptiMEM, according to manufacturer's instructions, for 24 h.

ON-TARGETplus Non-targeting Control Pool (D001810-10-05; Dharmacon, UK) was used as a transfection control. Following transfection, cells were serum-starved for a further 24 hours before being treated with HDM (100 µg/ml, CITEQ Biologics, The Netherlands). After 24 h exposure to HDM, cells were washed with sterile PBS and immediately lysed for RNA extraction (Monarch® Lysis Buffer; New England Biolabs, UK).

### qRT-PCR
Total RNA extraction was performed with the Monarch® Total RNA Miniprep Kit (New England BioLabs, UK), according to the manufacturer's instructions. RNA quality and quantity were assessed using a NanoDrop One Spectrophotometer (Thermofisher Scientific, UK). 1 ng/µl RNA was reverse transcribed to cDNA using a High-Capacity cDNA Reverse Transcription Kit (Applied Biosystems, UK) and Thermocycler (Bio-Rad, UK), according to manufacturer's instructions. RTqPCR was performed using CFX96 Real-Time PCR Detection Systems with CFX Manager analysis software (Bio-Rad, UK). Primers and TaqMan Fast Advanced Master Mix were obtained from Thermofisher Scientific (*YWHAZ* – Hs01122445_g1; *HMOX1* - Hs01110250 _m1*; MT1F* - Hs00232306_m1*; MT1M* - Hs00828387_g1; *MT2A* - Hs02379661_g1). Reactions and cycling conditions were as per the manufacturers' specifications. Fold change in gene expression was calculated using the $2^{-\Delta\Delta Ct}$ method.

### Immunofluorescence microscopy of frozen tissue sections
Snap-frozen skin samples were embedded in OCT (CellPath) and cut to 5–10 µm cryosections onto APES-coated slides. Sections were fixed in 4% paraformaldehyde, washed with PBS, blocked with PBS + 1% BSA + 10% FBS and incubated for 30 min with primary antibodies to the following markers: Langerin (Leica), multi-cytokeratin (Leica), CD3 (Dako), CD4 (Abcam), CD8 (Abcam), IL17 (rabbit polyclonal IgG, Abcam) or TNF (rabbit monoclonal IgG (clone D1G2), Cell Signalling Technology). After washing off the primary antibodies, secondary antibodies were added; these included: Alexa Fluor 488 goat anti-mouse IgG1a, Alexa Fluor 555 goat anti-rabbit IgG, and Alexa Fluor 647 goat anti-mouse IgG2b (all from ThermoFisher Scientific). Sections were then counterstained with DAPI (Sigma), mounted with Mowiol (Harco), coverslipped and imaged using an Olympus Dotslide scanning fluorescence microscope and Olympus VS-Desktop software. Antibody details are listed in Supplementary Data 14.

### RNA-seq
RNA was isolated using Direct-zol RNA micro prep (Zymo, UK) as per the manufacturer's protocol. RNA concentration and integrity were determined with an Agilent Bioanalyser (Agilent Technologies, Santa Clara, CA. Preparation of RNA-seq libraries and sequencing were carried out by Source Bioscience, UK. cDNA libraries were generated using SMART-Seq Stranded Library Preparation for Ultra Low Input according to the SMART-Seq Stranded Kit User Manual following the Ultra-low input workflow (Takara Bio). Samples were pooled (12/batch) for library preparation. Amplified libraries were validated on the Agilent BioAnalyzer 2100 to check the size distribution and on the Qubit High Sensitivity to check the concentration of the libraries. All the libraries passed the QC step. Sequencing was done on Illumina HiSeq 4000 instrument, 75 bp PE runs, $20 \times 10^6$ reads per sample.

### Drop-seq
Freshly dissociated whole skin biopsies were suspended in RNAse-out buffer and processed on ice to the co-encapsulation of single cells with genetically encoded beads (Drop-seq[58]). Monodisperse droplets at 1 nl in size were generated using the microfluidic devices fabricated in the Centre for Hybrid Biodevices, University of Southampton. To achieve single cell/single bead encapsulation with barcoded Bead SeqB (Chemgenes, USA), microfluidics parameters (pump flow speeds for

cells and bead inlets, cell buoyancy) were adjusted to optimise cell-bead encapsulation and the generation of high-quality cDNA libraries. Based on encapsulation frequencies and bead counts, up to 2000 STAMPS/samples were taken further for library prep (High Sensitivity DNA Assay, Agilent Bioanalyser, 12 peaks with the average fragment size 500 bp). The resulting libraries were run on a shared NextSeq run ($4 \times 10^4$ reads/cell for maximal coverage) at the Wessex Investigational Sciences Hub laboratory, University of Southampton, to obtain single-cell sequencing data

### Constellation-seq
Single cell libraries were generated using the Chromium Single Cell 3′ library and gel bead kit v3.1 from 10× Genomics. Briefly, cell suspensions were tagged using TotalSeq™ hashtag antibodies (Biolegend, Supplementary Data 14). TotalSeq-A anti-human Hashtag Antibody used at 0.5 µg (1 µL). After pooling, 10,000 viable cells were loaded onto a channel of the 10× chip to produce Gel Bead-in-Emulsions (GEMs). This underwent reverse transcription to barcode RNA before clean-up and cDNA amplification. cDNA was used for targeted linear amplification comprising 20 rounds of linear amplification (60 °C) using a pool of primers (Supplementary Data 7) at 40 nM and 0.4 µM of a P5 3′blocked primer as described previously[26]. cDNA libraries were purified twice using AMPure XP (Beckman Coulter) magnetic beads (1:0.6), and libraries were assessed using a Bioanalyser before tagmentation and Next-seq sequencing on an Illumina Nextseq500 (paired-end 28 × 60 bp reads).

### Bulk RNA-seq data analysis
Quality control of FASTQ files with raw sequence data was done using the FASTQC tool [FastQC: a quality control tool for high throughput sequence data. Available online at: http://www.bioinformatics.babraham.ac.uk/projects/fastqc]. High-quality reads filtered at 15 M depth across all samples were mapped to the human genome (GRCh38) using Kallisto[59]. Raw counts from RNA-Seq were processed in Bioconductor package EdgeR[60] and SLEUTH[61], the variance was estimated, and the size factor was normalised using the trimmed mean of *M*-values (TMM). Genes with a minimum of 2 read at a minimum of 50% samples were included in the downstream analyses. Differentially expressed genes (DEG) we identified applying significance threshold with false discovery rate (FDR) adjusted $p < 0.05$, |LogFC| >1. Normalised reads were taken for transcript-to-transcript co-expression analysis (BioLayout[62]). Pearson correlation coefficient r = 0.85, Markov Clustering Algorithm = 1.7. WGCNA analysis [37]were run on 5000 genes with maximum median absolute deviation (MAD) detected across LC transcriptomes from reactive and non-reactive patients from control and HDM patch tests at power = 4 module size 30 in R v 4.0.3 using voom transformed (TMM) normalised expression data post-QC checks. The summary profile (eigengene) for each module was correlated with external traits using the Pearson coefficient. Gene ontology analysis across clusters and modules was done using ToppGene online tool[63]. The protein interaction network was reconstructed from the top 50 genes with negative GS scores (reversely correlated with CD3 infiltration, Supplementary Data 12) in a turquoise module using STRING V11.5[64], a database of protein interactions, using default parameters. Gene Set Enrichment Analysis[36] was run for gene signatures identified in cross-talk analysis (Supplementary Data 9), setting contrasts for responders vs non-responding patients. For analysis of metallothionein gene profiles data from GSE150672 and GSE36287 were used.

### scRNA-seq data and Constellation-seq analysis
Single-cell RNA-seq and Constellation-seq analysis was carried out using pipelines established in Systems Immunology Group[26,33]. Following demultiplexing, raw FASTQ files were aligned to the human genome (GRCh38) using Kallisto-bustools[59]. CellBender[65] was used to

remove empty technical artefacts. Doublet detection and hashing demultiplexing were done using Solo[66]. Visualisation and clustering of scRNAseq was performed in Scanpy[27] following standard quality checks (empty barcodes, percentage of mitochondrial genes). Clusters were determined via single-cell neighbourhood analyses on the first principal components, followed by clustering and cell type identification (Leiden-based clustering[67]. Cell types were annotated based on the expression of known marker genes and cross-validated using publicly available single-cell transcriptomes[28]. Differentially regulated transcriptional networks were identified using model-based analysis of single-cell transcriptomics MAST[68]. Specific transcriptional signatures were tracked using Gene Set Expression Analysis[36,69]. Transcription factor activity prediction was done using DoRothEA[70]. Cell–cell communication was inferred using CelphoneDB[71] and CosstalkR[72].

## Whole exome data generation
Whole exome sequencing was performed by Macrogen, with data uploaded to the University of Southampton supercomputer Iridis5 in February 2021. Agilent SureSelect Human All Exon V6 capture kit was used for all 28 samples.

## Whole exome data analysis
The raw fastq files were aligned to the human genome reference GRCh38 with additional HLA regions included. Alignment was performed using BWA-MEM v0.7.15-r1140 and Samtools[73] v1.3.1. Picard (http://broadinstitute.github.io/picard/) was used to mark duplicates, sort the BAM files, index, and fix the mate pairs. GATK[74] v4 base quality score recalibration (BQSR) was used to detect systematic errors by the sequencing machine when it estimates the accuracy of each base call. Joint calling was executed using a bespoke script. GATK GenomicsDBImport was used to create a database of all g.vcf files for joint-calling and was targeted to the intersection between Agilent SureSelect V6 and Agilent SureSelect V5, both with ±150 bp padding. GATK Genotype GVCF has applied to joint-call the 28 AD samples with 1100 inflammatory bowel disease patients, also targeted to the intersection between Agilent SureSelect V6 and Agilent SureSelect V5 ±150 bp padding. A final script applied GATK Variant Quality Score Recalibration (VQSR), a technique applied on the variant callset that uses machine learning to model the technical profile of variants in a training set and uses that to flag probable artefacts from the callset. Annotation was completed using Ensembl VEP[75] v103. The joint-called vcf was uploaded to a local installation of seqr (https://github.com/broadinstitute/seqr) on a virtual machine for data visualisation, analysis, filtering and reporting. A suite of bespoke scripts was used to assess quality control. Bedtools[76] v2.26.0 was applied to calculate exome data coverage relative to the target capture kit. The GATK VariantEval tool was used to calculate various quality control metrics, including the number of raw or filtered SNPs and the ratio of transitions to transversions. These metrics are further stratified by functional class, CpG site, and amino acid degeneracy. Picard CollectVariantCallingMetrics was applied to collect the per-sample and aggregate (spanning all samples) metrics from the provided vcf file. Peddy (https://github.com/brentp/peddy) was executed locally in a Python conda environment to assess the relatedness of individuals and predict their ancestry and sex.

All QC metrics data were compiled into a Shiny App using R v 4.2.0 for data visualisation.

## Identifying filaggrin variants
We selected all loss of function filaggrin (FLG) variants identified from the cohort of 28 AD samples. We applied a maximum allele frequency (across all populations in gnomAD) of <0.05 and further filtered the variants in the AD cohort using an allele balance of >0.15 and genotype quality > 0.3 with VQSR applied as a flag.

## Applying GenePy to identify genes enriched in reactive vs. non-reactive patients
We assessed the difference in gene mutation burden between patients non-reactive and reactive to HDM using GenePy 1.3 (Mossotto et al.). We initially calculated GenePy scores for a pre-selected 34 genes (Supplementary Data 13) and compared their scores between non-reactive and reactive patients using logistic regression. We extrapolated this analysis to 2002 autoimmune genes (Supplementary Data 13) and compared GenePy scores using a Mann–Whitney-$U$ test.

## Statistics and reproducibility
The experimental group size was determined to match the most restrictive requirement: patient number for RNA-Seq analysis. Power calculations were done using "RNASeqPower" package in Bioconductor, R, 0.18129/B9.bioc.RNASeqPower, based on the preliminary data measuring expression levels of key molecular hubs in the LC gene regulatory network after exposure to epidermal cytokines[33]. To detect a statistically significant effect in a case–control experiment, with 20 million reads sequencing depth, experimental variation cv = 0.5, $\alpha = 0.01$, and 11 biological replicates per group provide >85% power to detect a 2-fold difference in gene expression levels. Power calculations for the flow cytometry analysis for specific marker: To detect a statistically significant difference ($\alpha = 0.05$, power > 80%) of 2-fold difference, a sample size of 3 is sufficient. The experiment group was not randomised. All patients were exposed to the control, and HDM patch test, and the responsiveness to HDM defined patient status. All statistical analyses were carried out using GraphPad Prism V9.2.0 unless specifically stated otherwise. Data distribution was tested for normality using the Kolmogorov–Smirnov test. Statistical significance was assessed by Mann–Whitney $U$-test for not-normally and $t$-test for normally distributed data as detailed across papers.

## Reporting summary
Further information on research design is available in the Nature Portfolio Reporting Summary linked to this article.

# Data availability
Sequencing data for RNA-seq and scRNA-seq is stored in the Gene Expression Omnibus database, accession number: GSE184509. The exome sequencing data underlying this article cannot be shared publicly due to ethical considerations. Source data are provided with this paper.

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

## Acknowledgements

We acknowledge the use of the Iridis5 High-Performance Computing Facility and Flow Cytometry Core Facilities, together with support services at the University of Southampton. The authors wish to acknowledge Nikki Graham, Senior Technician within the DNA laboratory, Dr Carolann McGuire and Dr Richard Jewell from Flow Cytometry Core for technical support. We are grateful to the nurses and administrative staff at the Clinical Research Facility, NIHR, University Hospital Southampton NHS Foundation Trust. We express our deepest gratitude to the patients recruited to the study. This research was funded in whole by the Wellcome Trust [Sir Henry Dale Fellowship 109377/Z/15/Z, awarded to MEP]. The 10× Chromium Controller was funded by a Cancer Research UK Advanced Clinician Scientist Fellowship to Sean Hua Lim (A27179). For the purpose of open access, the author has applied a CC BY public copyright licence to any Author Accepted Manuscript version arising from this submission. ML is funded by a BBSRC David Phillips Fellowship [BB/V004573/1], LSND was supported by an NIHR Southampton BRC Research Fellowship.

## Author contributions

M.E.P., M.A.J. and H.S.: intellectually conceived the study; E.C., Y.T., S.H., G.R., N.H., M.A.J., R.A. and M.E.P.: patient recruitment and clinical data acquisition; S.S., A.V., L.D., M.L., R.A., C.L. and G.D.: carried out experiments; S.S., A.V., M.E.P. and K.C.: carried out the analysis and meta-analysis of bulk RNA-seq data; S.S., A.V., M.E.P., K.C. and J.D.: carried out the analysis and meta-analysis of single-cell RNA-seq data; E.S. and S.E.: carried out WES data analysis; S.S., A.V., M.E.P.: wrote the paper; M.A.J., P.F., H.S., E.H., C.L.B.: discussed, and reviewed the paper.

## Competing interests

The authors declare no competing interests. M.E.P. started employment at Janssen Pharmaceutical Companies of Johnson & Johnson during the revision cycle of the paper. Janssen, or any of the employees/stakeholders have not been involved in any part or aspect of the project or paper.
