## [Peer Review File · Nature Communications]

Impaired expression of metallothioneins contributes to Th17/
TNF- mediated, allergen-induced inflammation in patients
with atopic dermatitis.REVIEWER COMMENTS

Reviewer #1 (Remarks to the Author):

The authors investigated responses to HDM patch testing in a relatively large group of patients (n=28) for a molecular study, allowing in vivo investigations in humans, on an atopic dermatitis background. Although eczematous reactions to HDM extracts will only reflect immune mechanisms of a subset of AD patients, it is an elegant model to study the actual human disease, using cutting edge techniques. The authors investigate non-reactive, irritant and reactive lesions, thus capturing the full range of clinical reactions to HDM exposure. They use HDM extracts, which is a common antigen used for atopy patch testing (although it is currently not commercially available). Comparable studies using single cell analyses have not previously been performed in this setting, and the information given has the potential to significantly advance the understanding of the phenomenon of atopy patch testing, and allergic reaction of the skin in general. I only have minor comments.

Minor

Why this strong bias towards T cells and LC, especially in flow cytometric studies? Was this a pragmatic decision? What about other immune cell population such as mast cells, potentially B cells, and monocytes/macrophages? Why were they not captured by flow cytometry?

Were those 28 patients selected in an unbiased manner, and the reactive vs nonreactive vs irritant populations reflect the average reaction pattern in a general AD population?

Reviewer #2 (Remarks to the Author):

The manuscript by Sirvent et al. employs an in vivo human allergen challenge, exposing patients with atopic dermatitis (AD) to house dust mite (HDM), to study immunotolerance vs inflammation in human skin. They report that responses to HDM are associated with activation of Langerhans cells (LCs) and T cells and increased baseline level of TNF. On the contrary, enhanced expression of metallothioneins in HDM non-reactive patients, protect against T cell mediated inflammation. These results are potentially of considerable interest for the field as they may pave the way to novel therapeutic strategies to limit unwanted inflammation and immune responses. The experimental approach is most interesting and highly translational. The bioinformatic work is extensive and impressive. It is a shame, however that the study plan did not include a cohort of healthy donors, nor a "true" baseline sample, taken prior to the challenge. Moreover, there are a number of issues associated with the flow cytometry immunophenotyping and the absolute need to perform functional experiments to validate the findings obtained through the bioinformatic analysis.

Major points:

The terms baseline and controls are used interchangeably e.g lines 205, 207). However, the only correct term is control and not baseline as the sample has not been taken prior to the HDM challenge. As immune cells travel to the site of the challenge, it is really not reasonable to assume that the immediately adjacent skin (which is where the control patch appear to have been placed, as per Fig 1b) is truly representative of unchallenged, baseline skin. Therefore, the use of baseline should be discontinued any relative interpretation of the data should refer to control.

Flow cytometry data in Figure 2,3, 4 are the weakest of the entire manuscript, as they raise a few questions of robustness and reliability. Besides some presentation issues that can be easily rectified [full representative gating strategy must be shown as supplementary figure;

exclude doublets while gating, axis should be fully labelled; clarify whether samples were acquired simultaneously or in batches and if the latter which steps were taken (e.g. application settings) to ensure reproducibility of data acquired over a certain period of time, add representative plots for data shown in Fig 3C and 4J)], regrettably the authors have not included a fixable live/dead staining step in their protocol (lines 955=968). As the skin biopsies have been digested at 37C ON a significant degree of cell death is to be expected and is well known to impact on subsequent staining and data reliability. In fact, all representative plots in Fig 2 show an unequivocal diagonal staining pattern at the tip of the double negative quadrant (most evident in 2H, I where they are discounted and so less relevant but still showing their presence), indicative of dead cells included in the final gating. Due to the lack of appropriate labelling, it is not clear what is on the x and y axes in Fig 2E,F. Assuming that CD3 is on the x axis, the staining is very dim, with no clear separation from the negative population, making hard to robustly quantify differences in cell frequencies among groups. In the HDM sample in 2F there is a further typical diagonal strike (top right corner), suggestive of antibody aggregates which may have also affected CD3+ cell quantification. This reviewer couldn't guess what was on the y axis in 2E,F, but whatever that may be, there is a clear difference in staining pattern and MFI in the two panels, suggestive of Reactive and Non-reactive samples been acquired in separate batches, possibly run without any standardization procedure (i.e. application settings). Similar issue also extends to data shown in Figure 3D, as MFI cannot be reliably compared across different experimental batches in absence of application settings. To overcome these multiple issues, this reviewer suggests making more extensive use of in situ immunophenotyping of cell populations of interest using immunofluorescence staining of existing patient slides to quantify changes in CD3+, LC, DDC, IL-17+ cells in a more reliable manner.

By performing extensive bioinformatics of RNA-Sequencing data the authors report that HDM reactivity is mediated by crosstalk between LC and TNF signalling and speculate in the discussion that TNF may impact on their LC maturation and or migration. Moreover, they go on to identify metallothionein genes to be enhanced in non-reactive patients, suggestive of a protective role against inflammatory responses to HDM. Functional experiments corroborating and validating these bioinformatic findings are needed, especially for the metallothionein as correlation between increased levels and non-reactive status is not causation.

Statistical analysis applied throughout the manuscript does not take in account that three group are being analysed (IRR, NON and REAC) and therefore t test or non-parametric Mann-Whitney should not be used. Instead ANOVA or non-parametric Kruskal-Wallis should be used, followed by the appropriate post hoc test. In some cases (Fig 2A, 3B), the legend doesn't say which test has been applied. Moreover, the Statistical analysis section (lines 1105-1106) should be adequately expanded to include information about how the normality of the sample has been determined and thus the decision to apply a parametric or a non-parametric test been made.

Samples number appear inconsistent throughout Figure 2. Panel D, G and J have only 11 patients in the NON group and panel G has only 9 in the REAC group., and consequently Panel K has 26 Reasons for excluding these samples should be outlined in the Statistical Analysis section of the manuscript. Also, the legend should list n numbers in the 3 groups for each panel.

The genetic analysis of the disease gene FLG is of interested however, it is not surprising that no statistical differences were observed, given the minimal sample size. A retrospective power calculation should be added to put this result in context. Moreover, the filtering criteria described in the Methods (lines 1093-1095) are not entirely clear. Does the allele balance and genotype quality reported refer to the data in gnomAD or to the current sample cohort? If the latter, why does the main text (line 187) have an allele balance of 0.19? Finally, are the 3 additional variants (line 186) novel or have they been reported before? Please also define "high quality" for these variants and include raw sequencing data as supplementary figure.

Minor points:

-Gene's name should be in italics

-TNF is just TNF, without alpha, which was used in an outdated nomenclature

REVIEWER COMMENTS

Reviewer #1 (Remarks to the Author):

The authors investigated responses to HDM patch testing in a relatively large group of patients (n=28) for a molecular study, allowing in vivo investigations in humans, on an atopic dermatitis background. Although eczematous reactions to HDM extracts will only reflect immune mechanisms of a subset of AD patients, it is an elegant model to study the actual human disease, using cutting edge techniques. The authors investigate non-reactive, irritant and reactive lesions, thus capturing the full range of clinical reactions to HDM exposure. They use HDM extracts, which is a common antigen used for atopy patch testing (although it is currently not commercially available). Comparable studies using single cell analyses have not previously been performed in this setting, and the information given has the potential to significantly advance the understanding of the phenomenon of atopy patch testing, and allergic reaction of the skin in general. I only have minor comments.

We thank the reviewer for assessing our study as comprehensive and interesting, noting in particular our ability to examine the transcriptional programming in human *in vivo* challenge setting.

Minor

Why this strong bias towards T cells and LC, especially in flow cytometric studies? Was this a pragmatic decision? What about other immune cell population such as mast cells, potentially B cells, and monocytes/macrophages? Why were they not captured by flow cytometry?

The bias in flow cytometry was caused by the study focus on the skin immune compartment, and the aim to investigate the function of Langerhans cells in the context of inflamed epidermis. The decision was pragmatic, and the design limited by the size of the biopsy, cellular content (with expected frequency of Langerhans cells at ~0.5%). By application of fluorescence – based LC sorting we were able to investigate their transcriptome in detail (which would be compromised if direct analysis in the whole cell population in biopsy was carried out, due to the low frequency). By combining the flow-cytometry focused approach to LCs and T cells, and unbiased single cell analysis of the complete cellular population we attempted to capture the totality of interactions in the skin.

Were those 28 patients selected in an unbiased manner, and the reactive vs nonreactive vs irritant populations reflect the average reaction pattern in a general AD population?

The 28 patients were recruited to the study accordingly to the study protocol, specifying active moderate to severe disease. Otherwise, there was no bias in the patient recruitment. Even though study numbers are limited (n=28), this is likely to be representative to the population of AD patients.

Reviewer #2 (Remarks to the Author):

The manuscript by Sirvent et al. employs an *in vivo* human allergen challenge, exposing patients with atopic dermatitis (AD) to house dust mite (HDM), to study immunotolerance vs inflammation in human skin. They report that responses to HDM are associated with activation of Langerhans cells (LCs) and T cells and increased baseline level of TNF. On the contrary, enhanced expression of metallothioneins in HDM non-reactive patients, protect against T cell mediated inflammation. These results are potentially of considerable interest for the field as they may pave the way to novel therapeutic strategies to limit unwanted inflammation and immune responses. The experimental approach is most interesting and highly translational. The bioinformatic work is extensive and impressive. It is a shame, however that the study plan did not include a cohort of healthy donors, nor a “true” baseline sample, taken prior to the challenge. Moreover, there are a number of issues associated with the flow cytometry immunophenotyping and the absolute need to perform functional experiments to validate the findings obtained through the bioinformatic analysis.

We thank the reviewer for assessing our study as potentially of considerable interest for the field, and for commenting on its usefulness for development of novel therapeutics in AD. While we agree that comparison with healthy skin can provide useful information about which processes are altered by the disease state. Such studies were indeed undertaken by other groups (e.g. Reynolds et al Science 2021, He et al JACI 2020). We aimed however to assess the responses to stimulation with an allergen, which would be absent in healthy skin. Such study design has an added benefit of design with internal control, allowing to minimise signal noise and biological variability.

Owing to its physiological function and biochemical structure, skin is one of the most challenging tissues to liberate the cells from. All the methods have been carefully optimised in the blood and healthy skin (gating for live and dead cells included in the Supplementary Figure 3), and all the samples processed in uniform way, including identical conditions for acquisition to enhance reproducibility.

“the absolute need to perform functional experiments to validate the findings obtained through the bioinformatic analysis”.

While we agree with the need for validation of the transcriptomics findings, it is important to stress, that the analysis we have performed are following a functional experiment (an *in vivo* challenge), which in itself is the most appropriate experimental system to test *in situ* responses to allergen. The model complexity is not possible to recapitulate *in vitro*, and would require a dedicated clinical trial, which is beyond the scope of the study. While the original study design included a number of cross-validation points (sequential analysis by flow cytometry and scRNA-seq, validation of key populations *in situ* by immune fluorescence, cross-validation of key identified programmes in transcriptomic data from skin disease and *in vitro* stimulated populations available in public domain), we have additionally:

- 1) Confirmed that the exposure to HDM allergen induces antioxidant responses in fibroblasts
- 2) Conducted siRNA silencing of *MTF1*, a transcription factor controlling the expression of metallothioneins,
- 3) Validated the effect of *MTF1* silencing on the expression of metallothioneins and *HMOX1*, a key molecule involved in regulation of responses to oxidative stress.
- 4) Performed Th17 staining *in situ* by immunofluorescence

Major points:

The terms baseline and controls are used interchangeably e.g lines 205, 207). However, the only

correct term is control and not baseline as the sample has not been taken prior to the HDM challenge.

We agree that the correct term is indeed “control”; the term “baseline” referring to the control patch test have been replaced throughout the manuscript.

As immune cells travel to the site of the challenge, it is really not reasonable to assume that the immediately adjacent skin (which is where the control patch appear to have been placed, as per Fig 1b) is truly representative of unchallenged, baseline skin. Therefore, the use of baseline should be discontinued any relative interpretation of the data should refer to control.

We thank the reviewer for that comment. We agree with the need to consider cell migration in the skin, and the term “baseline” have been replaced as “control site/control samples”. We would like to highlight, that the patch test side have been chosen to create the appropriate control for the challenge, as the skin composition, structure, microbiome, and pH change dramatically between body sites. At the same time, it was important to keep consistency between patients, to be able to draw most informative comparisons. The recruited patients had moderate to severe disease, and often presented with large areas of active eczema, further reducing where the patch test could be applied. It was also imperative to make this procedure acceptable to patients, and as least painful as possible.

Flow cytometry data in Figure 2,3, 4 are the weakest of the entire manuscript, as they raise a few questions of robustness and reliability. Besides some presentation issues that can be easily rectified [full representative gating strategy must be shown as supplementary figure; exclude doublets while gating, axis should be fully labelled; clarify whether samples were acquired simultaneously or in batches and if the latter which steps were taken (e.g. application settings) to ensure reproducibility of data acquired over a certain period of time, add representative plots for data shown in Fig 3C and 4J)],

The requested representative graphs have been added (Figure 3E, Figure 4K, Supplementary Figure 3A-E, Supplementary Figure 4D).

We would like to highlight, that while liberation of cells from the skin is challenging, and the resultant flow cytometry plots differ to pure PBMC preparations, we had optimised all the methods carefully in both blood and healthy skin (gating for live and dead cells included in the Supplementary Figure 3). Additionally, our isolation procedure was optimised for liberating maximum number of cells, from a small size biopsy, for multiple assays, including a reach and complex population of keratinocytes, as demonstrated in scRNA-seq data.

To enhance reproducibility all the samples were processed in a uniform way, including identical set-up of the instrument parameters for acquisition. The used cytometer setup includes internal controls to ensure that the laser intensity remains constant throughout, compensating for laser intensity decay over time and ensuring/increasing MFI comparability. For flow cytometry samples were processed immediately after extraction to avoid freezing of the tissue. Control and HDM patch test from same patient were always simultaneously processed.

regrettably the authors have not included a fixable live/dead staining step in their protocol (lines 955=968). As the skin biopsies have been digested at 37C ON a significant degree of cell death is to be expected and is well known to impact on subsequent staining and data reliability. In fact, all representative plots in Fig 2 show an unequivocal diagonal staining pattern

at the tip of the double negative quadrant (most evident in 2H, I where they are discounted and so less relevant but still showing their presence), indicative of dead cells included in the final gating.

We agree that inclusion of a live-dead stain to all the samples would be helpful, however, this was a pragmatic decision, due to the numbers of available colours in the FACS sorter. We agree with the statement, controls of FSC/SSC and singlet stain was included to limit this effect, and all the samples include internal matched control to ensure comparability. We performed digestion of healthy epidermis and PBMC stained with live/dead dye as control to ensure LC and T cells survive liberase digestion (Supplementary Figure 3). Viability of digested and sorted LC was as well checked (Supplementary Figure 3). Considering the viability percentages as acceptable we decided to run all samples applying those conditions.

As presented on the representative graphs, the staining for LCs indicates a clear, unequivocal population. We ensured that LCs, which were sorted directly for microbulk bioinformatic analyses were highly viable at the end of the digestion process, and the viability of all other cells was investigated in the bioinformatic analysis of scRNA-seq, where filtering for doublets/cells expressing high content of mitochondrial genes. The only feature investigated via flow cytometry was the cell composition, which was subsequently validated by scRNA-seq and visualised by IF.

Due to the lack of appropriate labelling, it is not clear what is on the x and y axes in Fig 2E,F.

This has now been corrected.

Assuming that CD3 is on the x axis, the staining is very dim, with no clear separation from the negative population, making hard to robustly quantify differences in cell frequencies among groups.

The isolation procedure reduces the intensity of staining for CD3, in comparison to T lymphocytes in the blood (Supplementary Figure 3). However, all the care was taken, as outlined above, to make the results comparable within the same patient and across the patient cohort, and the findings were subsequently validated by scRNA-seq and visualised by immunofluorescence.

In the HDM sample in 2F there is a further typical diagonal strike (top right corner), suggestive of antibody aggregates which may have also affected CD3+ cell quantification. This reviewer couldn't guess what was on the y axis in 2E,F, but whatever that may be, there is a clear difference in staining pattern and MFI in the two panels, suggestive of Reactive and Non-reactive samples been acquired in separate batches, possibly run without any standardization procedure (i.e. application settings).

We thank the reviewer for this comment, and we have now re-validated the flow cytometry to reduce the artefacts. We have gated the samples trying to avoid the diagonal staining as much as possible, keeping the same gating in Control and HDM for each patient.

Similar issue also extends to data shown in Figure 3D, as MFI cannot be reliably compared across different experimental batches in absence of application settings. To overcome these multiple issues, this reviewer suggests making more extensive use of in situ immunophenotyping of cell populations of interest using immunofluorescence staining of existing patient slides to quantify changes in CD3+, LC, DDC, IL-17+ cells in a more reliable manner.

To enable comparisons across the study we have used the same machine with optimised settings, for the duration of the study. Importantly, we report fold change in expression of CD3 and CD207 within each patient, setting up the gates on the relevant non-fluorescent control (Supplementary Figure 3) which overcomes the inter-sample/batch instability of MFI.

We thank the reviewer for the suggestion of identifying IL17 expressing T cells in situ. Excitingly, as shown in Figure 4L, these cells co-localise within LC:Tcell hubs. We regret, but due to the study design, only limited number of samples was available for the immunofluorescence, restricting our ability to quantify the changes in situ.

By performing extensive bioinformatics of RNA-Sequencing data the authors report that HDM reactivity is mediated by crosstalk between LC and TNF signalling and speculate in the discussion that TNF may impact on their LC maturation and or migration. Moreover, they go on to identify metallothionein genes to be enhanced in non-reactive patients, suggestive of a protective role against inflammatory responses to HDM. Functional experiments corroborating and validating these bioinformatic findings are needed, especially for the metallothionein as correlation between increased levels and non-reactive status is not causation.

Our past and recent work documents extensively the effect of TNF on human LCs, both in vitro and upon migration across activation, migration and stimulatory properties (Polak et al 2012, Polak et al 2014, Polak et al 2017, Sirvent 2020, Polak and Singh 2021). Similar findings have been already reported by others (Cumberbatch et al, Br J Dermatol 1999, Cumberbatch et al, Clin Exp Immunol 2003, Epaulard et al J Immunol, 2014, De la Cruz Diaz JID Innovations 2021). The study presented here indicates that the TNF action described by us previously has a critical importance at the patch test site in situ, in the skin of patients with AD, upon exposure to HDM. The complexity of the system is too great to replicate in vitro, and the only way to further corroborate it in humans would be through a clinical trial, which is beyond a scope of this study, and might indeed be an exciting next step.

We thank the reviewer for suggesting functional studies addressing the role of metallothioneins in skin responses to HDM stimulation. We have confirmed in an in vitro model system, that exposure to HDM drove expression of *HMOX1*, a key molecule in the responses to oxidative stress. Silencing of *MTF1*, a transcription factor regulating the expression of metallothioneins, reduced both the expression of metallothioneins, and HDM-induced *HMOX1*, providing the causal link between allergen exposure, anti-oxidant responses, and the protective role of metallothioneins. The data is presented in Figure 6I, and Supplementary Figure 6D,E.

Statistical analysis applied throughout the manuscript does not take in account that three groups are being analysed (IRR, NON and REAC) and therefore t test or non-parametric Mann-Whitney should not be used. Instead ANOVA or non-parametric Kruskal-Wallis should be used, followed by the appropriate post hoc test.

We thank the reviewer for pointing out the need to include the group of IRR in the statistical comparisons. As the three cohorts of patients, irritant, nonreactive and reactive were independent, which precludes ANOVA analysis, we have added the results of t-tests to the graphs.

In some cases (Fig 2A, 3B), the legend doesn't say which test has been applied. Moreover, the Statistical analysis section (lines 1105-1106) should be adequately expanded to include information about how the normality of the sample has been determined and thus the decision to apply a parametric or a non-parametric test been made.

We thank the reviewer for bringing up this point. Both statistical analysis section (Lines 1161-1263) and in figure legends.

Samples number appear inconsistent throughout Figure 2. Panel D, G and J have only 11 patients in the NON group and panel G has only 9 in the REAC group., and consequently Panel K has 26 Reasons for excluding these samples should be outlined in the Statistical Analysis section of the manuscript. Also, the legend should list n numbers in the 3 groups for each panel.

We thank the reviewer for highlighting the need of explanations. We recruited 28 patients. One of the samples has been excluded due to disagreement on the PT outcome (observed minimal redness, but not consistent with the patch test perimeter). FACS results for CD3 T cells were compromised for one sample due to a technical fault. One sample has been processed for single cell RNA-seq only. This explanation has been added to the method section (lines 683-688), and the sample numbers in each group added to the figure legend.

The genetic analysis of the disease gene FLG is of interested however, it is not surprising that no statistical differences were observed, given the minimal sample size. A retrospective power calculation should be added to put this result in context.

We agree the sample size was too small for assessment of FLG variants. We included these measurements given the importance of FLG status for AD, as one of the factors investigating skin barrier quality. We feel that adding a power calculation is of limited value when the sample size is clearly underpowered. We have made it explicit in the text (lines 189-190) that we were underpowered to detect statistical differences.

Moreover, the filtering criteria described in the Methods (lines 1093-1095) are not entirely clear. Does the allele balance and genotype quality reported refer to the data in gnomAD or to the current sample cohort?

We have amended the section on identifying fillagrin variants to make it explicit that the allele balance and genotype quality were applied to our sample cohort. We have also amended the typo regarding an allele balance, which should say >0.15 .

If the latter, why does the main text (line 187) have an allele balance of 0.19? Finally, are the 3 additional variants (line 186) novel or have they been reported before? Please also define "high quality" for these variants and include raw sequencing data as supplementary figure.

We apologise for this error, which was due to a typo when quoting the allele balance in the materials and methods section, which has been corrected to 0.15 (line 1149). Two of the variants are not novel, as in they are in population databases. One is novel in gnomAD. We have added the allele frequencies to Supplementary Table 2. We have also defined high quality (line 186-187).

Minor points:

-Gene's name should be in italics

All gene names have been italicised.

-TNF is just TNF, without alpha, which was used in an outdated nomenclature

We have replaced the term TNF α with TNF throughout the manuscript.

REVIEWER COMMENTS

Reviewer #1 (Remarks to the Author):

The authors have sufficiently responded to the reviewers' concerns.

Reviewer #2 (Remarks to the Author):

The authors have reasonably addressed the comments raised. Inconsistencies have been resolved and technical issues clarified and/or rectified. The addition of functional experiments about the role of metallothioneins in skin responses to HDM stimulation strengthens the study and validates the bioinformatic approach taken. Overall, this is an important piece of work that will advance the field.

I have only one residual comment and is about the statistical test used to analyse the 3 groups. In response to the comment that three group are being analysed (IRR, NON and REAC) and therefore and therefore t test or non-parametric Mann-Whitney should not be used, they state: "As the three cohorts of patients, irritant, nonreactive and reactive were independent, which precludes ANOVA analysis, we have added the results of t-tests to the graphs".

What do they mean by that? That the experiments were not performed at the same time and therefore they cannot be directly compared? In that case they should not be presented on the same graph. Comparison of more than two group does require ANOVA or related non-parametric test, followed by post hoc test.

Reviewer #3 (Remarks to the Author):

In their paper, "TNF signalling in the cutaneous immune network instructs local Th17 allergen-specific inflammatory responses in atopic dermatitis", Sirvent et al. generate an array of RNA sequencing, flow cytometry, imaging, and cytokine data in skin and blood biopsies from patients with atopic dermatitis under control and exposure to house dust mite (HDM) conditions. A wide range of analyses were carried out in identifying transcriptional and other omic and cellular changes that were identified between control and HDM skin exposed regions and between individuals who reacted to the HDM challenge, those who didn't react, and those who had general skin irritation as a result of the control and HDM patches. From these data the authors constructed a series of biological hypotheses relating to the role Th17 T cells in the skin play in explaining the inflammatory reaction that were observed in some of the patients who responded to the HDM skin challenge vs. those who did not.

I note that I was not a reviewer for the initial submission of this manuscript to Nature Communications.

Using innovative experimental designs to characterize complex immune conditions in skin such as AD should be the norm in science, so I do love this type of study where there is access to the relevant tissue under relevant conditions in a matched way. The authors have also employed state of the art molecular and cellular profiling techniques as well as analyses to wade through very high-dimension data, albeit with moderate sample sizes. On the one hand, the authors clearly demonstrate a number of informative hypotheses from the excellent data they generated. On the other, the authors also I believe overreach in terms of the claims given there is minimal prospective experiments carried out in support of the hypotheses they generated (as detailed more fully in the specific comments below). Further,

I note the paper is very dense, a lot of material, over 50 panels in the main figures along, with probably 7500 or more words in the intro, results, and discussion (not including long legends), so nontrivial to review. Perhaps is somewhat of an embarrassment of riches from the data generated, but I think the paper would benefit from the showing of fewer results in the main text, and then further fleshing out the more impactful results that support the primary claims the authors want to make in the paper (I have tried to speak to this in my specific comments below).

Finally, I note the authors did not do the reviewers any favors in making the material more accessible, where for example there were no labels on the files or in table legends for the supplementary tables, and so it was just quite annoying and more time consuming to have to use my special signal processing skills to match the contents of the xlsx and csv supplementary tables provided to the appropriate table tag in order to figure out what was being referred to and where. For example, the excel spreadsheet corresponding to supplementary table 1 looks like:

(see pdf of review I attached to this review for the screen grab)

Where you can see from this there is no indication that this is Supplementary Table 1, and then the name of this file in what I was able to pull down from the Nature Comm reviewer website was:

332180_1_related_ms_6814480_rgpc4h.xlsx

Where again you can see there is no way to discern that this is supplementary table 1.

Specific Comments:

1. Throughout the paper the authors make stronger claims in my view than are supported by the data. For example, the section starting at line 172 is entitled, "Reactivity to HDM is mediated by co-expansion of T cells and LCs", but what is shown in this section (figure 2) is correlative, so that there is an association between T cell and LC expansions and HDM; the data are support the claim as an hypothesis rather than a demonstrated causal association. Sometimes in the text the authors speak appropriately regarding the claims in contrast to the section headings. For example, in the above case, the others state in line 217 that the correlation observed is "...suggesting that immune crosstalk between these cells perpetuates the responses to allergen." In any case, if the authors want to make strong causal claims, then they need to do an experiment that demonstrates the causal relationship or point to other data/known results where the correlation they observe is causal.

2. Section title on lines 248 and 249, same comment as #1 comment.

3. For the WGCNA analysis carried out, it is not clear what the input was into that; were all samples included (control and HMD patched skin? Reactive and non-reactive and irritated all together? Etc.) If all combined, the pearson correlations used in this analysis are inflated given multiple samples from the same person which will be more correlated with each other than across individuals. In addition, identifying only 7 modules is a little bit of a red flag given generally you would expect more, so these would appear less specific (e.g., the grey "non-module" and turquoise module contained > 60% of the genes), more general and maybe indicating an issue with the parameters chosen. More details are needed to understand what this actually represents. Further, in the analysis of the modules, it is unclear what is being done. For example, line 286 indicates a correlation between the yellow module and EASI. What is being correlated? The first principal component vector of the yellow model and EASI? Some differential expression measure for genes in this module between control and HDM regions?

4. For the single cell analysis done in figure 3f-h and described in paragraph starting at line 289, there are only 3 individuals that were profiled, yet pretty sweeping statements are made. For example, line 296 states, "...in non-reactive patients, a network of epidermal signaling to LCSs...", but there is only 1 non-reactive individual profiled! And then claims about the differences then in the 2 reactive patients profiled and 1 non-reactive patient profiled couldn't possibly be supported with such small sample sizes. Either the same signal and observations being made in these patients needs to be generalized in all other patients (this is somewhat done in the paragraph starting at line 299) using the bulk profiling data (where, for example, deconvolution analysis could be carried out to break the bulk tissue up into cell types and then carry out analysis on those cell types with the increased sample size), or the hypothesis generated from the single cell data needs to be prospectively validated in a second group.

5. The results for figure 4 seem interesting, but again it is difficult to interpret what is shown because it is unclear what universe of transcription factors under the different conditions were considered. For example, in figure 4E there are 5 TFs shown that are upregulated in CR vs. CRN patients. How many TFs were examined? Why were these 5 chosen? Were there others that were not consistent with the biological narrative used in the main text around this figure? Were there other TFs that would be expected given the biology being claimed that were not detected? The p-values given in figures 4F and G do not look as though they are all significant at the 0.05 level and there appears to be no adjusted p values that are < 0.01 ; were these all of the pathways that were identified or were these selected given they are consistent with the biological narrative? Throughout the impression given is one of cherry-picking results that support the narrative the authors want to sell as opposed to a completely data driven uncovering of that narrative. I am not claiming the story being told is not completely data driven, only that it is not possible to discern with what is provided. I would think fewer results that are more fully fleshed out would make for a stronger paper. Perhaps there are too many riches coming from these data and the authors want to tell all, but the end result to this reviewer seems to be a long set of narratives where the results are not as well fleshed out or validated as they need to be to convince.

6. Related to the above point, the paragraph starting on line 373 talks about cytokine profiling carried out to support or refute hypotheses supported in the previous few paragraphs. This is the type of follow-on experimentation that is very useful in my view to carry out to build evidence for hypotheses generated from the transcriptomic data. I think the authors could focus more on a limited number of results and then use these follow-on assays and analyses to really drive their points home. As it stands, it is not clear what the primary hypothesis is being tested with the cytokine profiling experiment, but rather some additional results are given without all the appropriate contexts. What would be nice is to start this paragraph with an explicit indication about what one would expect to see with the cytokine profiling if the Th17 overexpression results identified from the transcriptomic analyses was real. What cytokine differences would you expect to see. What would you not expect to see. Explicitly state the hypothesis to be tested and then run the experiment and test it and report on those results.

7. For figure 5 results, I have no idea what FDR p means (see for example line 423). Do the others mean q value?

8. In the paragraph starting on line 409, the authors indicate they are going to determine the "...cellular and molecular drivers of active T cell...". They then go on to describe figure 5 results from single cell data that demonstrates a range of interesting differential expression patterns from the cell clusters identified from the scanpy analysis (I do like this kind of analysis), and identify genes like MAP4K5 that are over expressed in various cell types of interest. They then use that such genes have support in the literature as being drivers of

response to environmental stress. While these findings may be interesting, the authors certainly have not shown these as key driver genes of the activated T cell states they observe. These are hypotheses the authors are generating and that have some support in the literature. But without experimental validation, nothing in my view has been “determined” (so this is related to my comment #1 above).

9. Figure 6F the authors talk about a protein interaction analysis. I could not find any description on what exactly was done, what data were used to assemble this and so on. It is simply impossible to interpret what is being shown without knowing the details.

Reply to Reviewer comments

REVIEWER COMMENTS

Reviewer #1 (Remarks to the Author):

The authors have sufficiently responded to the reviewers' concerns.

We thank the Reviewer 1 for accepting our explanations.

Reviewer #2 (Remarks to the Author):

The authors have reasonably addressed the comments raised. Inconsistencies have been resolved and technical issues clarified and/or rectified. The addition of functional experiments about the role of metallothioneins in skin responses to HDM stimulation strengthens the study and validates the bioinformatic approach taken. Overall, this is an important piece of work that will advance the field.

I have only one residual comment and is about the statistical test used to analyse the 3 groups. In response to the comment that three group are being analysed (IRR, NON and REAC) and therefore and therefore t test or non-parametric Mann-Whitney should not be used, they state: "As the three cohorts of patients, irritant, nonreactive and reactive were independent, which precludes ANOVA analysis, we have added the results of t-tests to the graphs".

What do they mean by that? That the experiments were not performed at the same time and therefore they cannot be directly compared? In that case they should not be presented on the same graph. Comparison of more than two group does require ANOVA or related non-parametric test, followed by post hoc test.

We thank the Reviewer2 for accepting our explanations. We have adjusted the statistical test for this analysis to ANOVA.

Reviewer #3 (Remarks to the Author):

In their paper, "TNF signalling in the cutaneous immune network instructs local Th17 allergen-specific inflammatory responses in atopic dermatitis", Sirvent et al. generate an array of RNA sequencing, flow cytometry, imaging, and cytokine data in skin and blood biopsies from patients with atopic dermatitis under control and exposure to house dust mite (HDM) conditions. A wide range of analyses were carried out in identifying transcriptional and other omic and cellular changes that were identified between control and HDM skin exposed regions and between individuals who reacted to the HDM challenge, those who didn't react, and those who had general skin irritation as a result of the control and HDM patches. From these data the authors constructed a series of biological hypotheses relating to the role Th17 T cells in the skin play in explaining the inflammatory reaction that were observed in some of the patients who responded to the HDM skin challenge vs. those who did not.

I note that I was not a reviewer for the initial submission of this manuscript to Nature Communications.

Using innovative experimental designs to characterize complex immune conditions in skin such as AD

should be the norm in science, so I do love this type of study where there is access to the relevant tissue under relevant conditions in a matched way. The authors have also employed state of the art molecular and cellular profiling techniques as well as analyses to wade through very high-dimensional data, albeit with moderate sample sizes. On the one hand, the authors clearly demonstrate a number of informative hypotheses from the excellent data they generated.

We thank Reviewer 3 for appreciating the breadth of the study, quality of generated data, and for commenting on the range of analysis carried out and highlighting the state of the art molecular and cellular profiling.

On the other, the authors also I believe overreach in terms of the claims given there is minimal prospective experiments carried out in support of the hypotheses they generated (as detailed more fully in the specific comments below). Further, I note the paper is very dense, a lot of material, over 50 panels in the main figures along, with probably 7500 or more words in the intro, results, and discussion (not including long legends), so nontrivial to review. Perhaps is somewhat of an embarrassment of riches from the data generated, but I think the paper would benefit from the showing of fewer results in the main text, and then further fleshing out the more impactful results that support the primary claims the authors want to make in the paper (I have tried to speak to this in my specific comments below).

We agree with Review 3 on the complexity of the findings, and challenges in presentation they cause. Having carried out unique extensive unbiased analysis of the data from our *in vivo* human challenge model (for the first time in such system, to the best of our knowledge) we aimed to share the unbiased full picture of changes happening at the site of the challenge. We think that the volume of submitted text and data is in line with other manuscripts recently published in Nature Communications, based on next generation sequencing approaches, and describing new aspects of biology in human tissues (<https://www.nature.com/articles/s41467-022-34975-2>, <https://www.nature.com/articles/s41467-022-33202-2>).

However, we agree that the way we presented the data in Figure 5 detracted from the main message of the manuscript. We have combined all the Constellation seq analysis in Figure 3, and kept the focus on the T cell compartment in the main body of the manuscript. The length of the manuscript body now is reduced to 4615 words, with 5 main and 5 supplementary figures. We have also re-worded the text carefully, to avoid overinterpretation of findings, and we focused the report on the Th17/TNF signalling and changes in metallothionein expression, two aspects of biology where the most significant changes were observed, and which had been addressed experimentally. We feel the suggested changes significantly improve the clarity of the manuscript, and we are grateful for these recommendations.

Finally, I note the authors did not do the reviewers any favors in making the material more accessible, where for example there were no labels on the files or in table legends for the supplementary tables, and so it was just quite annoying and more time consuming to have to use my special signal processing skills to match the contents of the xlsx and csv supplementary tables provided to the appropriate table tag in order to figure out what was being referred to and where. For example, the excel spreadsheet corresponding to supplementary table 1 looks like:

(see pdf of review I attached to this review for the screen grab)

Where you can see from this there is no indication that this is Supplementary Table 1, and then the name of this file in what I was able to pull down from the Nature Comm reviewer website was:

332180_1_related_ms_6814480_rgpc4h.xlsx

Where again you can see there is no way to discern that this is supplementary table 1.

We apologise for the problems with formatting, of which we were not aware. We believed that the submitted tables were entitled appropriately (e.g. Supplementary Table 1), and the submission check-list did not indicate these titles were re-formatted by the system. To avoid the problem on the re-submission, all table titles have now been added in the file content.

Specific Comments:

1. Throughout the paper the authors make stronger claims in my view than are supported by the data. For example, the section starting at line 172 is entitled, "Reactivity to HDM is mediated by co-expansion of T cells and LCs", but what is shown in this section (figure 2) is correlative, so that there is an association between T cell and LC expansions and HDM; the data do not support the claim as an hypothesis rather than a demonstrated causal association. Sometimes in the text the authors speak appropriately regarding the claims in contrast to the section headings. For example, in the above case, the authors state in line 217 that the correlation observed is "...suggesting that immune crosstalk between these cells perpetuates the responses to allergen." In any case, if the authors want to make strong causal claims, then they need to do an experiment that demonstrates the causal relationship or point to other data/known results where the correlation they observe is causal.

We thank the Reviewer for highlighting this issue. We have re-worked the manuscript text to ensure any overstatements have been removed throughout. We have reworded section titles and legend titles to reflect only the findings and avoid the inclusion of possible implications/interpretation. For example, the section title that the Reviewer highlighted now reads: "Reactivity to HDM is associated with co-expansion of T cells and LCs." Line 165

2. Section title on lines 248 and 249, same comment as #1 comment.

This section title has been replaced with: "Expression of metallothioneins counterbalances LC overactivation differentiating HDM reactive and non-reactive patients" (lines 318-319)

3. For the WGCNA analysis carried out, it is not clear what the input was into that; were all samples included (control and HDM patched skin? Reactive and non-reactive and irritated all together? Etc.) If all combined, the Pearson correlations used in this analysis are inflated given multiple samples from the same person which will be more correlated with each other than across individuals. In addition, identifying only 7 modules is a little bit of a red flag given generally you would expect more, so these would appear less specific (e.g., the grey "non-module" and turquoise module contained > 60% of the genes), more general and maybe indicating an issue with the parameters chosen. More details are needed to understand what this actually represents. Further, in the analysis of the modules, it is unclear what is being done. For example, line 286 indicates a correlation between the yellow module and EASI. What is being correlated? The first principal component vector of the yellow module and EASI? Some differential expression measure for genes in this module between control and HDM regions?

For WGCNA analysis samples from reactive and non-reactive patients were included, both control and HDM stimulated. The motivation for that approach was to discover modules correlating with clinical features within any given set of samples (control responders, control non-responders, HDM responders, HDM non-responders), as much as across all of the samples. Indeed, the package author, Peter Langfelder, advises to include all the samples in a paired experiment, as the

appropriate use of the WGCNA method (<https://support.bioconductor.org/p/86486/>). We agree that very few modules were identified, however, this is not unexpected for human primary Langerhans cells, which have a very stable transcriptome, with very few genes changing expression levels, on the stimulation, and the changes are usually moderate, in comparison with other types of dendritic cells, as documented by us in DOI: 10.1038/jid.2013.375 and in <https://www.nature.com/articles/s41467-019-14125-x>, and by others at DOI: 10.1189/jlb.1107750.

The correlation with clinical and laboratory-measured features external to the dataset has been computed within the standard WGCNA algorithm, correlating summary profile (eigengene) for each module, with external traits and identifying the most significant associations based on the Pearson coefficient. A more detailed explanation was added to the methods section, Supplementary material, lines 989-994.

4. For the single cell analysis done in in figure 3f-h and described in paragraph starting at line 289, there are only 3 individuals that were profiled, yet pretty sweeping statements are made. For example, line 296 states, "...in non-reactive patients, a network of epidermal signaling to LCSs...", but there is only 1 non-reactive individual profiled! And then claims about the differences then in the 2 reactive patients profiled and 1 non-reactive patient profiled couldn't possibly be supported with such small sample sizes. Either the same signal and observations being made in these patients needs to be generalized in all other patients (this is somewhat done in the paragraph starting at line 299) using the bulk profiling data (where, for example, deconvolution analysis could be carried out to break the bulk tissue up into cell types and then carry out analysis on those cell types with the increased sample size), or the hypothesis generated from the single cell data needs to be prospectively validated in a second group.

We thank the Reviewer for this suggestion. While the TNF-mediated cross-talk between T cell and APCs had been investigated previously in Constellation 10X data, supporting its importance for HDM responses (current Figure 3JK), we have now tracked the signature in LC bulk transcriptomes, and documented TNF protein expression by immunofluorescence. Deconvoluting Tcell:LC signalling cross-talk signal from bulk RNA-seq of LC isolated from skin biopsies of responding and non-responding patients (NR n = 11, R n=7) indeed strengthened the evidence for the stronger cross-talk between LC and T cells in responders. Additionally, we confirmed co-localisation of TNF protein and CD3 T cells in the skin of HDM-responding patients (Supplementary Figure 3M).

5. The results for figure 4 seem interesting, but again it is difficult to interpret what is shown because it is unclear what universe of transcription factors under the different conditions were considered. For example, in figure 4E there are 5 TFs shown that are upregulated in CR vs. CRN patients. How many TFs were examined? Why were these 5 chosen? Were there others that were not consistent with the biological narrative used in the main text around this figure? Were there other TFs that would be expected given the biology being claimed that were not detected? The pvalues given in figures 4F and G do not look as though they are all significant at the 0.05 level and there appears to be no adjusted p values that are < 0.01; were these all of the pathways that were identified or were these selected given they are consistent with the biological narrative?

All the analysis in Figure 3 A-E (previously Figure 4, T cells examined by constellation-seq) prior to focusing on Th17 signalling were carried out in an unbiased way, without selecting for any specific programmes or gene sets. Constellation-seq panel (Supplementary table 4) was enriched for 1161 transcripts (as mentioned in the legend for figure 3 (formerly legend for figure 4) encompassing immunology and skin relevant genes, including transcription factors, to limit the number of dropouts in genes with low abundance. Analysis of differential regulons in T cell responding and non-responding patients was run using Dorothea (<https://saezlab.github.io/dorothea/>), a benchmarked

integrated resource for the estimation of human transcription factor activities, against a complete set of 1541 human transcription factors in the tool. The transcription factors presented in Figure 3D are the 5 top as per Wilcoxon signed-rank test enriched in each patient category, and they fully supported the differences in activation status of responding vs non-responding patients. We have added the lines denoting significance to the graphs with gene ontology analysis, these were the top unbiased enriched categories, and these were the very few which reached statistical significance. We have also included Supplementary Table 6 listing the Gene Ontology Terms with corresponding statistical analysis.

Throughout the impression given is one of cherry-picking results that support the narrative the authors want to sell as opposed to a completely data driven uncovering of that narrative.

Throughout the manuscript the analysis has been done in a completely unbiased way, utilising the full set of detected genes, in bulk RNA-seq, Drop-seq, and Constellation-seq. Constellation-seq has been indeed focused on 1161 transcripts, relevant to immune cells and skin biology, to enrich for the signal relevant to the study versus sequencing of unrelated, highly abundant genes. Constellation-seq has been thoroughly validated by the group against unbiased sequencing of whole transcriptome, and we demonstrated that it did not introduce artefacts for the enriched genes (for reference: Vallejo et al 2021, <https://doi.org/10.1016/j.isci.2021.102147>). It was only after the key (the strongest) signal in the dataset was determined, that the hypothesis-driven analysis was conducted (including the analysis suggested by the Reviewer, e.g. tracking of TNF signalling signal in bulk RNA-seq).

I am not claiming the story being told is not completely data driven, only that it is not possible to discern with what is provided. I would think fewer results that are more fully fleshed out would make for a stronger paper. Perhaps there are too many riches coming from these data and the authors want to tell all, but the end result to this reviewer seems to be a long set of narratives where the results are not as well fleshed out or validated as they need to be to convince.

We have aimed to provide all the outcomes of the analyses carried out in the supplementary material, and indeed, the manuscript volume reflects the breadth of analysis carried out, as expected in the Next Generation sequencing high dimensional data. We regret that the problems with the accessibility of the supplementary material, as highlighted by the Reviewer 3, might have contributed to the impression of “cherry-picking” the data but can confirm that was not the case or intention. We trust that with the current revisions these reservations have been addressed, and that it is now transparent how the unbiased analysis led to selecting, following, and validating specific aspects of the biology.

6. Related to the above point, the paragraph starting on line 373 talks about cytokine profiling carried out to support or refute hypotheses supported in the previous few paragraphs. This is the type of follow-on experimentation that is very useful in my view to carry out to build evidence for hypotheses generated from the transcriptomic data. I think the authors could focus more on a limited number of results and then use these follow-on assays and analyses to really drive their points home. As it stands, it is not clear what the primary hypothesis is being tested with the cytokine profiling experiment, but rather some additional results are given without all the appropriate contexts. What would be nice is to start this paragraph with an explicit indication about what one would expect to see with the cytokine profiling if the Th17 overexpression results identified from the transcriptomic analyses was real. What cytokine differences would you expect to see. What would you not expect to see. Explicitly state the hypothesis to be tested and then run the experiment and test it and report on those results.

We thank the reviewer for this suggestion, and we feel that the re-structured manuscript has gained clarity and presents a stronger argument. We have specifically introduced this hypothesis testing component in lines 211-212, 283-285, 288-294, 322-324, 352-354, 407-409 to ensure that the reader understands the rationale for each of the subsequent experiments.

7. For figure 5 results, I have no idea what FDR p means (see for example line 423). Do the others mean q value?

We used the phrase “FDR p value” to denote FDR (False Discovery Rate) adjusted p value, (alternatively known as q value). To avoid confusion, we added the explanation to the figure legend and method section (line 986).

8. In the paragraph starting on line 409, the authors indicate they are going to determine the “...cellular and molecular drivers of active T cell...”. They then go on to describe figure 5 results from single cell data that demonstrates a range of interesting differential expression patterns from the cell clusters identified from the scanpy analysis (I do like this kind of analysis), and identify genes like MAP4K5 that are over expressed in various cell types of interest. They then use that such genes have support in the literature as being drivers of response to environmental stress. While these findings may be interesting, the authors certainly have not shown these as key driver genes of the activated T cell states they observe. These are hypotheses the authors are generating and that have some support in the literature. But without experimental validation, nothing in my view has been “determined” (so this is related to my comment #1 above).

We agree with the Reviewer that while interesting, and relevant to skin biology, the outcomes of the unbiased analysis of Constellation-seq data in the previous figure 5 are mainly hypothesis-generating without experimental validation. We aimed to describe the unbiased full picture of molecular changes observed at the sites of the patch test, and we presented the state of subclinical inflammation in the control, uninvolved skin of responding patients. To streamline the manuscript, and to keep it focused, Figure 5 has been now removed from the manuscript, and the summary of Constellation-seq analysis included in Supplementary Figure 3.

9. Figure 6F the authors talk about a protein interaction analysis. I could not find any description on what exactly was done, what data were used to assemble this and so on. It is simply impossible to interpret what is being shown without knowing the details.

An explanation detailing STRING analysis of protein interaction network has been added to the methods section, lines 995-1000.

REVIEWERS' COMMENTS

Reviewer #2 (Remarks to the Author):

none

Reviewer #3 (Remarks to the Author):

I have reviewed the revision of the manuscript entitled, "Impaired expression of metallothioneins contributes to Th17/TNF-mediated, allergen-induced inflammation in atopic dermatitis" by Sirvent et al., which had originally been titled, "TNF signalling in the cutaneous immune network instructs local Th17 allergen-specific inflammatory responses in atopic dermatitis". My main issue with the manuscript I had originally reviewed had been around the more sweeping claims that were being made and that I believed were not fully supported by the results presented. In addition, there was lack of clarity in several critical parts of the paper that made it difficult to assess exactly what was done and then whether what was done supported the claims made.

I believe the authors have adequately addressed the concerns I had raised, appropriately modifying the manuscript to bring claims into line with what the results show, and clarifying around others points I had made. I think the experimental design, data generated and analyses and interpretations provided will be of strong interest to the Nature Communications readership.